# Dissecting the phase separation and oligomerization activities of the carboxysome positioning protein McdB

Joseph L Basalla[1], Claudia A Mak[2], Jordan A Byrne[1], Maria Ghalmi[1], Y Hoang[1], Anthony G Vecchiarelli[1]*

[1]Department of Molecular, Cellular, and Developmental Biology, University of Michigan-Ann Arbor, Ann Arbor, United States; [2]Department of Biological Chemistry, University of Michigan-Ann Arbor, Ann Arbor, United States

**Abstract** Across bacteria, protein-based organelles called bacterial microcompartments (BMCs) encapsulate key enzymes to regulate their activities. The model BMC is the carboxysome that encapsulates enzymes for $CO_2$ fixation to increase efficiency and is found in many autotrophic bacteria, such as cyanobacteria. Despite their importance in the global carbon cycle, little is known about how carboxysomes are spatially regulated. We recently identified the two-factor system required for the maintenance of carboxysome distribution (McdAB). McdA drives the equal spacing of carboxysomes via interactions with McdB, which associates with carboxysomes. McdA is a ParA/ MinD ATPase, a protein family well studied in positioning diverse cellular structures in bacteria. However, the adaptor proteins like McdB that connect these ATPases to their cargos are extremely diverse. In fact, McdB represents a completely unstudied class of proteins. Despite the diversity, many adaptor proteins undergo phase separation, but functional roles remain unclear. Here, we define the domain architecture of McdB from the model cyanobacterium *Synechococcus elongatus* PCC 7942, and dissect its mode of biomolecular condensate formation. We identify an N-terminal intrinsically disordered region (IDR) that modulates condensate solubility, a central coiled-coil dimerizing domain that drives condensate formation, and a C-terminal domain that trimerizes McdB dimers and provides increased valency for condensate formation. We then identify critical basic residues in the IDR, which we mutate to glutamines to solubilize condensates. Finally, we find that a condensate-defective mutant of McdB has altered association with carboxysomes and influences carboxysome enzyme content. The results have broad implications for understanding spatial organization of BMCs and the molecular grammar of protein condensates.

*For correspondence: ave@umich.edu

Competing interest: The authors declare that no competing interests exist.

## Editor's evaluation

This work builds on the recent identification of the two-factor system that was discovered to be essential for the maintenance of carboxysome distribution (McdAB). McdB, a member of the two-factor system, is the focus of study here, and the intent is to uncover the driving forces for and the functional roles of phase separation. The key findings are that an N-terminal intrinsically disordered region modulates condensate solubility, a central coiled-coil dimerizing domain drives condensate formation, and oligomerization through the C-terminal domain provides the increased valency that contributes to the associative phase transitions.

**eLife digest** Cells contain many millions of protein molecules that must be in the right place at the right time to carry out their roles. A process called phase separation, in which a well-mixed solution separates into two phases – one concentrated and one dilute – is thought to help organize the contents of various cell types.

The single-celled bacteria *Synechococcus elongatus* converts carbon dioxide from the air into sugars using internal reaction centers. This process depends on a protein called McdB which is crucial for spatially organizing these centers. McdB readily phase separates on its own in a test tube, raising the possibility that this phenomenon could be involved in the carbon dioxide-capturing process.

To investigate, Basalla et al. identified the parts of McdB responsible for phase separation and modified them to make a version that was less able to separate. When viewed under the microscope, *Synechococcus elongatus* cells containing the altered McdB showed changes in the organization and structure of the reaction centers. This suggests that phase separation by McdB is required for optimal carbon capture by this bacterium.

In the future, manipulation of McdB phase separation could be used to improve carbon capture technologies or enhance crop growth. Phase separation is also known to influence complex disease. Therefore, further understanding of the process could be important for developing new disease treatments.

## Introduction

Compartmentalization is a fundamental feature by which cells regulate metabolism. Although bacteria lack extensive lipid–membrane systems, recent reports have shown that proteinaceous bacterial microcompartments (BMCs) are a widespread strategy for compartmentalization in bacteria (*Kerfeld et al., 2018*; *Sutter et al., 2021*). Briefly, BMCs are nanoscale reaction centers where key enzymes are encapsulated within a selectively permeable protein shell. The best studied BMC is the carboxysome, found within cyanobacteria and other autotrophic bacteria (*Kerfeld et al., 2018*; *Yeates et al., 2008*). Carboxysomes encapsulate the enzyme ribulose-1,5-bisphosphate carboxylase/oxygenase (Rubisco) with its substrate $CO_2$ to significantly increase the efficiency of carbon fixation. Carboxysomes serve as a paradigm for understanding BMC homeostasis, including assembly, maintenance, permeability, and spatial regulation (*Kerfeld et al., 2018*; *Sutter et al., 2021*; *Yeates et al., 2008*). Furthermore, to engineer efficient carbon-fixing organisms, efforts to express functional carboxysomes in heterologous hosts are ongoing (*Long et al., 2018*; *Flamholz et al., 2020*).

An important aspect of BMC homeostasis is spatial regulation (*Savage et al., 2010*). We recently identified the two-protein system responsible for spatially regulating carboxysomes, which we named the maintenance of carboxysome distribution (McdAB) system (*MacCready et al., 2018*; *MacCready et al., 2020*; *MacCready et al., 2021*). McdA and McdB function to prevent carboxysome aggregation, thereby ensuring optimal function and equal inheritance upon cell division (*MacCready et al., 2018*; *MacCready et al., 2021*). Briefly, McdA is an ATPase that forms dynamic gradients on the nucleoid in response to an adaptor protein, McdB, which associates with carboxysomes (*MacCready et al., 2018*; *MacCready et al., 2021*). The interplay between McdA gradients on the nucleoid and McdB-bound carboxysomes result in the equal spacing of carboxysomes down the cell length of rod-shaped bacteria. This mode of spatial regulation by McdA is typical for the widespread and well-studied ParA/MinD family of positioning ATPases, of which McdA is a member (*Hakim et al., 2021*). ParA/MinD ATPases spatially organize an array of genetic- and protein-based cargos in the cell, including plasmids, chromosomes, the divisome, flagella, and other mesoscale complexes (*Lutkenhaus, 2012*; *Vecchiarelli et al., 2012*). While ParA/MinD ATPases are highly similar in sequence and structure, the partner proteins that act as adaptors and link the ATPases to their respective cargo are highly diverse, largely due to partners providing cargo specificity. Indeed, McdB represents an entirely new class of adaptor proteins, and it is therefore unknown how McdB interacts with itself, McdA, and carboxysomes to confer specificity, or how these interactions are regulated. Bioinformatic analyses show that McdAB systems also exist for several other BMC types (*MacCready et al., 2021*). Therefore, an understanding of the biochemical properties of McdB, and how these properties influence its behavior in vivo, are

important next steps to advancing our knowledge on the spatial regulation of carboxysomes and BMCs in general.

From our initial studies in the model cyanobacterium *Synechococcus elongatus* PCC 7942 (*Se7942*), we found that McdB self-associates in vitro to form both a stable hexamer (*MacCready et al., 2021*) and liquid-like condensates (*MacCready et al., 2020*). However, the domain architecture of McdB and the regions required for its oligomerization and condensate formation are unknown. Our understanding of how proteins form biomolecular condensates has rapidly developed over the past decade. Briefly, biomolecular condensates are the result of molecules having demixed out of solution to form a dense, solvent-poor phase that exists in equilibrium with the soluble phase (*Banani et al., 2017*; *Alberti et al., 2019*; *Mittag and Pappu, 2022*). This process occurs under a specific set of conditions where protein–protein interactions are more favorable than protein–solvent interactions (*Alberti et al., 2019*). A biochemical understanding of how proteins form condensates in vitro has led to a deeper understanding of how this process facilitates subcellular organization in both eukaryotic and prokaryotic cells (*Dignon et al., 2020*; *Azaldegui et al., 2021*). Furthermore, characterizing the underlying chemistries for diverse biomolecular condensates has led to the development of these condensates as synthetic tools to engineer cytoplasmic organization (*Peeples and Rosen, 2021*; *Lasker et al., 2021*; *Viny and Levine, 2020*; *Shin et al., 2017*). Thus, a major focus of this report is to characterize the biochemistry of McdB, including its condensate formation, and link these properties to the spatial regulation of carboxysomes in vivo.

Here, we define a domain architecture of *Se7942* McdB, identify the domains contributing to oligomerization and condensate formation, and discover a potential interplay between these two modes of self-association. We then create a series of point mutations that allow us to fine-tune the solubility of McdB condensates both in vitro and in vivo without affecting McdB structure or oligomerization. Finally, we use this mutation set to identify in vivo phenotypes that relate specifically to the ability of McdB to form condensates and associate with carboxysomes. The findings have implications for the use of carboxysomes in synthetic biology approaches, designing biomolecular condensates, and general BMC biology.

## Results

### Structural predictions generate a low-confidence α-helical model for *Se7942* McdB

We first set out to determine the *Se7942* McdB crystal structure. However, McdB displayed robust phase separation across a range of buffer conditions, making crystal trials thus far unsuccessful (*Figure 1—figure supplement 1*). We next turned to I-TASSER (Iterative Threading ASSEmbly Refinement) (*Roy et al., 2010*; *Yang et al., 2015*) to generate structural models, which predicted the McdB secondary structure to be predominantly α-helical but with a disordered N-terminus (*Figure 1—figure supplement 2A*). I-TASSER also generates full-length atomic models of the target sequence that are consistent with its secondary structure predictions via multiple sequence alignments using top matches from the protein databank (PDB). The top 3 models were once again almost entirely α-helical, with the top model also showing a disordered N-terminus (*Figure 1—figure supplement 2B*). But ultimately, the top 10 PDB matches identified by I-TASSER aligned poorly with McdB, with each alignment showing low sequence identity (<20% on average) and low-quality scores (*Z*-scores <1 on average) (*Figure 1—figure supplement 2C*). As a result, the top 3 final models generated by I-TASSER all have poor confidence scores (*Figure 1—figure supplement 2B*). These findings are not surprising in context with our previous bioinformatic analyses showing that cyanobacterial McdBs are highly dissimilar to other characterized proteins at the sequence level, potentially related to the high disorder content of McdBs (*MacCready et al., 2020*; *Echave et al., 2016*). Together, these data provide low-confidence structural predictions for *Se7942* McdB. We therefore set out to validate these predictions with empirical approaches.

### Defining a tripartite domain architecture for *Se7942* McdB

We used circular dichroism (CD) to characterize the secondary structure of *Se7942* McdB. The spectrum showed a characteristic α-helical signature that remained stable even after incubation at 80°C (*Figure 1A*), indicative of a stabilized coiled-coil (CC; *Fiumara et al., 2010*). This is consistent with

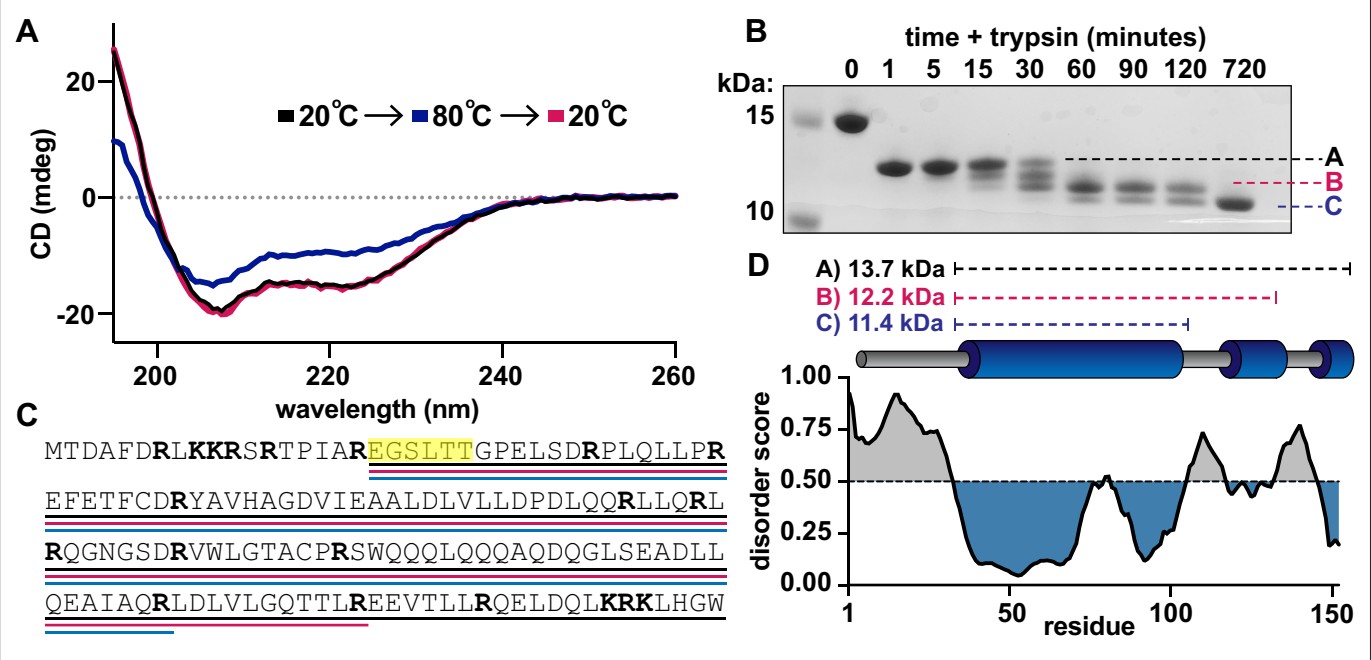

**Figure 1.** Defining a domain architecture of *Se7942* McdB. (**A**) Circular dichroism (CD) spectra of McdB at 20°C (black), 80°C (blue), and then returned to 20°C (magenta). Spectra show α-helical structure resilient to heat denaturation. (**B**) Sodium dodecyl sulfate–polyacrylamide gel electrophoresis (SDS–PAGE) analysis of trypsin-digested McdB sampled over digestion time. Bands labeled A, B, and C were isolated and N-terminally sequenced. (**C**) Amino acid sequence of McdB with basic residues (Lys-K and Arg-R) in bold. Regions corresponding to bands A, B, and C from panel B are underlined in black, magenta, and blue, respectively. Amino acids determined through N-terminal sequencing of bands A, B, and C are highlighted yellow. (**D**) Structural model of *Se7942* McdB. Regions corresponding to bands A, B, and C are indicated with predicted molecular weights (MWs) (*top*). Predicted secondary structure of McdB (*middle*) aligned with a Predictor of Natural Disorder Regions (PONDR) plot using the VLXT algorithm (*bottom*) with disordered regions colored gray and predicted α-helical domains in blue.

The online version of this article includes the following source data and figure supplement(s) for figure 1:

**Source data 1.** Sodium dodecyl sulfate–polyacrylamide gel electrophoresis (SDS–PAGE) gel corresponding to the trypsin digest gel in *Figure 1B*.

**Source data 2.** Spreadsheet containing the raw data for circular dichroism (CD) curves shown in *Figure 1A*.

**Figure supplement 1.** Phase separation of *Se7942* McdB across a range of buffer conditions during crystal screens.

**Figure supplement 2.** I-TASSER predictions for *Se7942* McdB.

**Figure supplement 2—source data 1.** Spreadsheet with an editable version of the table in *Figure 1—figure supplement 2C*.

the helical predictions from I-TASSER, and with our previous bioinformatics data that predicted CC domains to be conserved across all cyanobacterial McdB homologs (*MacCready et al., 2020*).

We next sought to empirically identify folded domains using limited proteolysis (*Laureto et al., 2008*; *Schopper et al., 2017*; *Fontana et al., 2004*; *Figure 1B*). Trypsin cuts at arginines and lysines, which are frequent throughout *Se7942* McdB – the largest fragment between any two basic residues is ~3 kDa (*Figure 1C*). Therefore, any stably folded regions that are protected from trypsin would be resolved via this approach. The digestion yielded three major bands of varying stabilities, which we labeled A, B, and C (*Figure 1B*). B and C were most stable, representing an ~11 kDa fragment that remained undigested for 12 hr. This strong protection against trypsin is consistent with our CD data, which showed high resilience to heat denaturation (see *Figure 1A*).

We next subjected bands A, B, and C to N-terminal sequencing to determine the location of these stably folded regions in McdB. All three bands had the same N-terminal sequence starting at E19 (*Figure 1C*). Therefore, the first 18 amino acids at the N-terminus were digested within the first minute to produce band A, and further digestion progressed slowly from the C-terminus to produce bands B and C (*Figure 1B, C*). By combining the N-terminal sequencing results (*Figure 1C*), the molecular weights of the three protected regions (*Figure 1B*), the locations of all arginines and lysines (*Figure 1C*), and the predicted disorder via PONDR VLXT (*Xue et al., 2019*; *Figure 1D*), we

developed a model for the domain architecture of *Se7942* McdB that was consistent with I-TASSER predictions (*Figure 1D*).

## *Se7942* McdB forms a trimer-of-dimers hexamer

From our structural model, we defined three major domains of *Se7942* McdB: (1) an intrinsically disordered region (IDR) at the N-terminus, (2) a highly stable central CC, and (3) a C-terminal domain (CTD) with two short helical regions. We used this model to design a series of truncation mutants, including each of these domains alone as well as the CC domain with either the N-terminal IDR or the CTD (*Figure 2A*). CD spectra of these truncations showed that the N-terminus was indeed disordered on its own, and both the CC domain and CTD maintained α-helical signatures (*Figure 2B*, *Figure 2—figure supplement 1A*).

Having previously shown that full-length *Se7942* McdB forms a hexamer in solution (*MacCready et al., 2021*), we used these truncations to determine which domains contributed to oligomerization. We first ran size-exclusion chromatography (SEC) with each McdB truncation, and used the full-length protein as a reference for where the hexamer elutes. Although the molecular weight of each monomeric truncation is within ~5 kDa of one another (*Figure 2—figure supplement 1B*), we found that only the CC + CTD construct eluted at a volume similar to the full-length hexamer (*Figure 2—figure supplement 1C*). Furthermore, the CC domain alone or with the IDR, appeared to elute between the expected monomer and hexamer peaks, suggesting that the CC domain with or without the IDR forms an oligomeric species that is smaller than a hexamer.

To further resolve the oligomeric states of these McdB truncations, we performed size-exclusion chromatography coupled to multi-angled light scattering (SEC-MALS) on each of the constructs that appeared to oligomerize during SEC (*Figure 2C*). We found that the CC + CTD truncation was indeed hexameric, while the CC domain, with or without the IDR, formed a dimer (*Figure 2D*). These data suggest that the CC domain of McdB contains a dimerization interface, while the CTD subsequently allows for trimerization of dimers. Although we were unable to generate an atomic level structure of McdB nor determine the orientation of monomers within the hexamer, we have identified two key oligomerizing domains and conclude that full-length *Se7942* McdB forms a hexamer as a trimer-of-dimers.

## *Se7942* McdB forms condensates via pH-dependent phase separation coupled to percolation

Recent reports on the formation of protein condensates have begun to unveil an interplay between phase separation and network formation or 'percolation' (*Kar et al., 2022*; *Mittag and Pappu, 2022*). These reports have shown that, instead of forming strictly through liquid–liquid phase separation, many proteins undergo phase separation coupled to percolation (PSCP) to form condensates (*Kar et al., 2022*; *Mittag and Pappu, 2022*). The definitions of phase separation, percolation, and PSCP are rigorous and nuanced. But broadly speaking, phase separation can often involve incompatibilities in solubility to drive transitions in density, while percolation concerns multivalent interactions to form dense networks (*Mittag and Pappu, 2022*). For protein systems undergoing PSCP, different regions of the protein can facilitate solubility than the regions that facilitate multivalent networking (*Mittag and Pappu, 2022*). We therefore set out to identify if full-length McdB showed signs of PSCP to guide our investigation on how the different domains of McdB affect condensate formation and stability.

Evidence for PSCP has recently been shown by studying the time-dependent viscoelastic nature of condensates (*Harmon et al., 2017*; *Wang et al., 2018a*; *Mittag and Pappu, 2022*) and the formation of networks at subsaturating concentrations (*Kar et al., 2022*; *Mittag and Pappu, 2022*). We therefore first used fluorescence recovery after photobleaching (FRAP) to assess the viscoelastic nature of McdB condensates (*Alberti et al., 2019*). Newly formed condensates recovered within minutes (*Figure 3A*), and readily fused and relaxed into spheres within seconds (*Figure 3B*), reminiscent of liquid-like fluid droplets. But after 18 hr, recovery was significantly slower (*Figure 3A*) and these 'mature' droplets no longer fused. The time-dependent changes in material properties of McdB condensates are a signature of PSCP (*Mittag and Pappu, 2022*).

Next, we determined a saturation concentration ($c_{sat}$) for McdB condensate formation in our standard buffer conditions (100 mM KCl, 20 mM 4-(2-hydroxyethyl)-1-piperazineethanesulfonic acid (HEPES), pH 7.2). Condensates were observed at or above 2 μM, suggesting a $c_{sat}$ between 1 and 2 μM

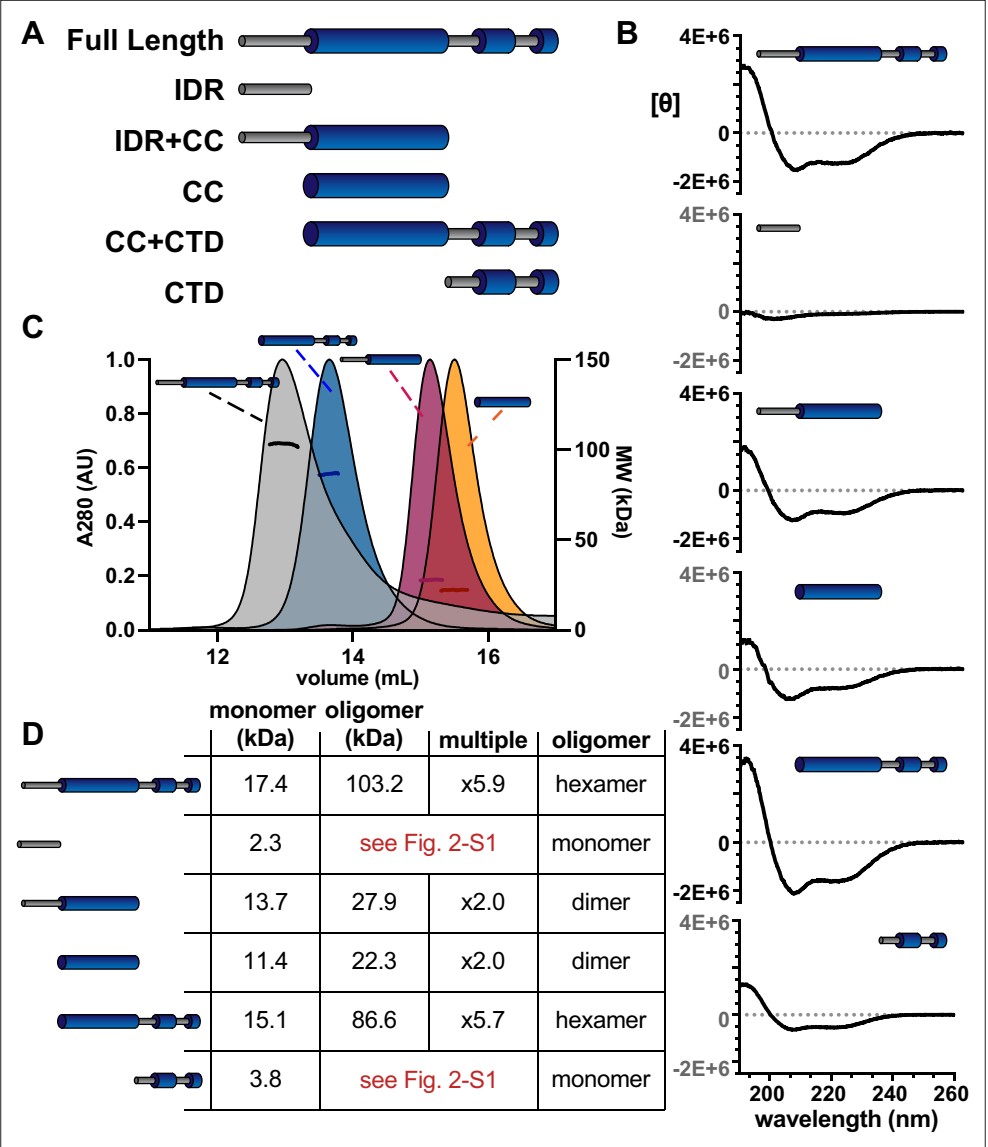

**Figure 2.** The α-helical domains of McdB form a trimer-of-dimers hexamer. (**A**) Illustration of McdB truncations generated based on the predicted domain structure. (**B**) Circular dichroism (CD) spectra normalized by MW for the indicated McdB truncations. Spectra show α-helical content for all truncations, except for the disordered N-terminal fragment. (**C**) Size-exclusion chromatography coupled to multi-angle light scattering (SEC-MALS) for full-length McdB and truncation mutants that showed oligomerization activity (see *Figure 2—figure supplement 1*). (**D**) Summary of the SEC-MALS data from (**C**).

The online version of this article includes the following source data and figure supplement(s) for figure 2:

**Source data 1.** Spreadsheet containing the raw data for circular dichroism (CD) curves shown in *Figure 2B*.

**Source data 2.** Spreadsheet containing the raw data for size-exclusion chromatography coupled to multi-angle light scattering (SEC-MALS) curves shown in *Figure 2C* and an editable version of the table in *Figure 2D*.

**Figure supplement 1.** McdB truncations have unique secondary structures and display different oligomeric states.

**Figure supplement 1—source data 1.** Sodium dodecyl sulfate–polyacrylamide gel electrophoresis (SDS–PAGE) gel corresponding to the McdB truncation mutants versus the His-SUMO tag as described in *Figure 2—figure supplement 1B*.

**Figure supplement 1—source data 2.** Spreadsheet containing the raw data for circular dichroism (CD) curves shown in *Figure 2—figure supplement 1A*.

**Figure supplement 1—source data 3.** Spreadsheet containing the raw data for SEC curves shown in *Figure 2—figure supplement 1C*.

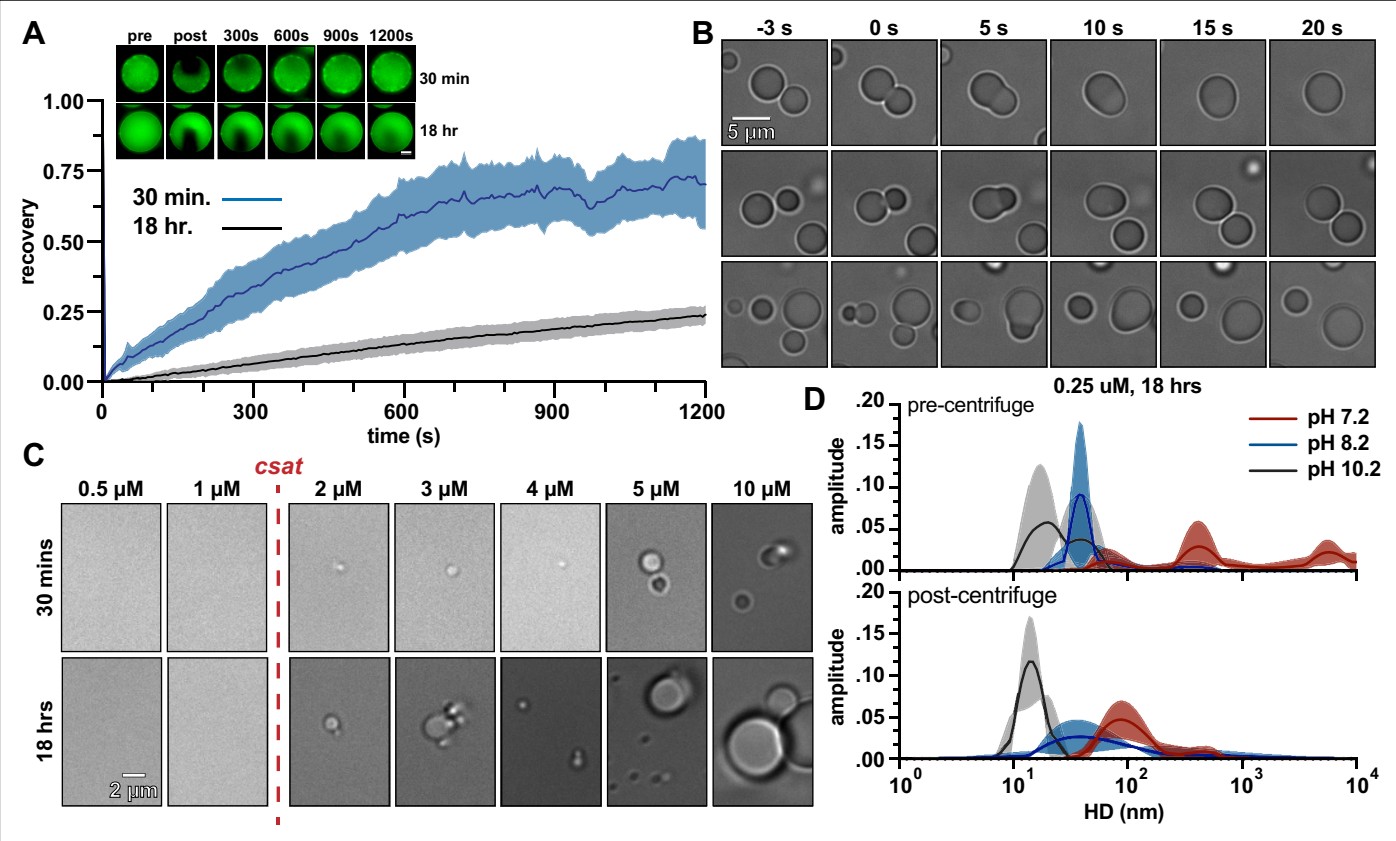

**Figure 3.** McdB from *Se7942* forms liquid-like condensates via pH-dependent phase separation coupled to percolation (PSCP). (**A**) Fluorescence recovery after photobleaching (FRAP) of McdB condensates at the indicated time points. Means and standard deviation (SD) from $n = 8$ condensates are shown. Representative fluorescence microscopy images for condensates incubated at 30 min or 18 hr are shown (inlet, scale bar = 2 µm). (**B**) Representative differential interference contrast (DIC) microscopy timeseries showing newly formed McdB condensates fusing and relaxing into spheres on the order of seconds. Scale bar applies to all images. (**C**) Representative DIC microscopy images at the indicated protein concentrations. McdB condensates were seen at and above concentrations of 2 µM, suggesting a saturation concentration ($c_{sat}$) between 1 and 2 µM. Scale bar applies to all images. (**D**) Dynamic light scattering (DLS) of McdB at a concentration ~1/10 the $c_{sat}$ determined from (**C**) and at increasing pH values as indicated. Samples were analyzed both before (*top*) and after (*below*) a 5 min spin at 20,000 × *g*. Larger 'networks' are seen forming at lower pHs, even below the observed $c_{sat}$.

The online version of this article includes the following source data for figure 3:

**Source data 1.** Spreadsheet containing the raw data for fluorescence recovery after photobleaching (FRAP) curves shown in *Figure 3A*.

**Source data 2.** Spreadsheet containing the raw data for dynamic light scattering (DLS) curves shown in *Figure 3D*.

(*Figure 3C*). We then used dynamic light scattering to investigate whether McdB formed network-like species at concentrations below $c_{sat}$ as done previously for other proteins (*Kar et al., 2022*). At 0.25 µM (~1/10 the observed $c_{sat}$) in our standard buffer (pH 7.2), McdB displayed a heterogeneous size distribution of defined species spanning 100–1000 nm in diameter (*Figure 3D*); significantly larger than a monodispersed hexamer (*Stetefeld et al., 2016*). Even after high-speed centrifugation, McdB in the supernatant remained mainly as mesoscale clusters on the order of 100 nm, suggesting the formation of McdB clusters at pH 7.2 (*Figure 3D*).

Our previous work has shown that McdB condensates are solubilized at higher pH values (*MacCready et al., 2020*), therefore we set out to determine if McdB clusters remained in solution at higher pH. Interestingly, the clusters were pH dependent. At pH 8.2, the largest clusters (>100 nm) were lost, but smaller clusters (~60 nm) remained. At pH 10.2, a single homogeneous species remained at ~10 nm (*Figure 3D*), which is consistent with the hydrodynamic diameter of a monodispersed McdB hexamer (*Stetefeld et al., 2016*). Together, the data show that McdB can form mesoscale clusters at sub-saturation concentrations and that McdB condensates show time-dependent changes to visco-elasticity, suggesting McdB forms condensates via pH-dependent PSCP.

## The CC domain of *Se7942* McdB is necessary and sufficient for condensate formation

We next used the truncations to determine how each domain of *Se7942* McdB affects condensate formation and stability. Interestingly, no McdB truncations formed condensates under the buffer conditions that sufficed for full-length McdB (*Figure 4A*). This finding suggests that no single domain of McdB is sufficient for full-length level condensate formation. Rather, all domains must influence McdB condensates to some extent.

When we added a crowding agent (10% PEG) to increase the local protein concentration, both the IDR and CTD alone were unable to form condensates, even at concentrations up to 4 mM (*Figure 4B*). However, all truncations containing the CC domain formed condensates. In fact, the CC domain alone was necessary and sufficient for forming condensates, albeit at much higher concentrations than full-length McdB. McdB condensates formed and fused similarly in the presence of other crowding agents, showing these activities were not PEG specific (*Figure 4—figure supplement 1*). It should be noted that we did not observe condensates forming by any truncation in the absence of a crowding agent, leaving open the possibility that these agents are functioning as multivalent co-assemblers and not simply crowding agents. Still, the data show that the CC domain is necessary for condensate formation and thus implicates this domain as the driver of McdB condensate formation.

## The IDR and CTD domains are modulators of McdB condensate formation

Although the IDR and CTD were not required for the CC domain to form condensates, fusing either of these domains back onto the CC increased condensate formation and size (*Figure 4B*). By using centrifugation to quantify the amount of protein in the dense versus light phases (*Alberti et al., 2019*), we found that the addition of either the IDR or CTD onto the CC domain comparably increased condensate formation (*Figure 4C*). However, by performing FRAP on 'mature' condensates (incubated for 18 hr), we found that the IDR + CC condensates recovered much faster than all other constructs, including full-length McdB (*Figure 4D*). Moreover, newly formed IDR + CC condensates fused and relaxed into spheres an order of magnitude faster than newly formed full-length McdB condensates (*Figure 4E*).

Together, we draw the following conclusions: (1) The CC domain is necessary and sufficient for condensate formation, although to a lesser extent than full-length McdB; (2) The IDR increases solvent interactions, thus affecting phase separation of the CC domain, but seemingly not percolation. This is supported by the fact that the IDR + CC construct has increased condensate formation compared to the CC alone. However, IDR + CC condensates do not mature, lacking the change in viscoelasticity seen for full-length McdB (see *Figure 3A*); (3) the CTD increases multivalent interactions to support condensate formation via PSCP. This occurs ostensibly by increasing oligomerization, which in turn increases valency (*Bracha et al., 2018*; *Guillén-Boixet et al., 2020*), and by the CTD itself providing network-forming contacts within condensates; and lastly (4) the fact that full-length McdB, which contains the IDR and CTD, does not show the same fluid-like behavior as the IDR + CC (no CTD), suggests that the CTD may interact with the IDR within condensates formed by full-length McdB. Using this information, we next sought out residues that affect condensate formation, but not McdB structure or its ability to form a hexamer.

## Net charge of the IDR modulates McdB condensate solubility

To determine which types of residues influence condensate formation, we first performed turbidity assays across a range of protein concentrations, salt concentrations, and pH as previously described (*Mitrea et al., 2018*; *Alberti et al., 2019*). Over all McdB concentrations, the phase diagrams showed decreased turbidity at higher KCl concentrations (*Figure 5—figure supplement 1*), implicating electrostatic interactions. We also found that turbidity decreased at higher pH (*Figure 5—figure supplement 1*), suggesting that positively charged residues are important in the solubilization of condensates. We used centrifugation to quantify the amount of McdB in the dense versus light phases across KCl and pH titrations while keeping McdB concentration constant. Again, we found a clear increase in the soluble fraction and decreases in condensate size and number as both KCl (*Figure 5A*) and pH was increased (*Figure 5B*). The data reveal a critical role for positively charged residues in McdB condensate stability.

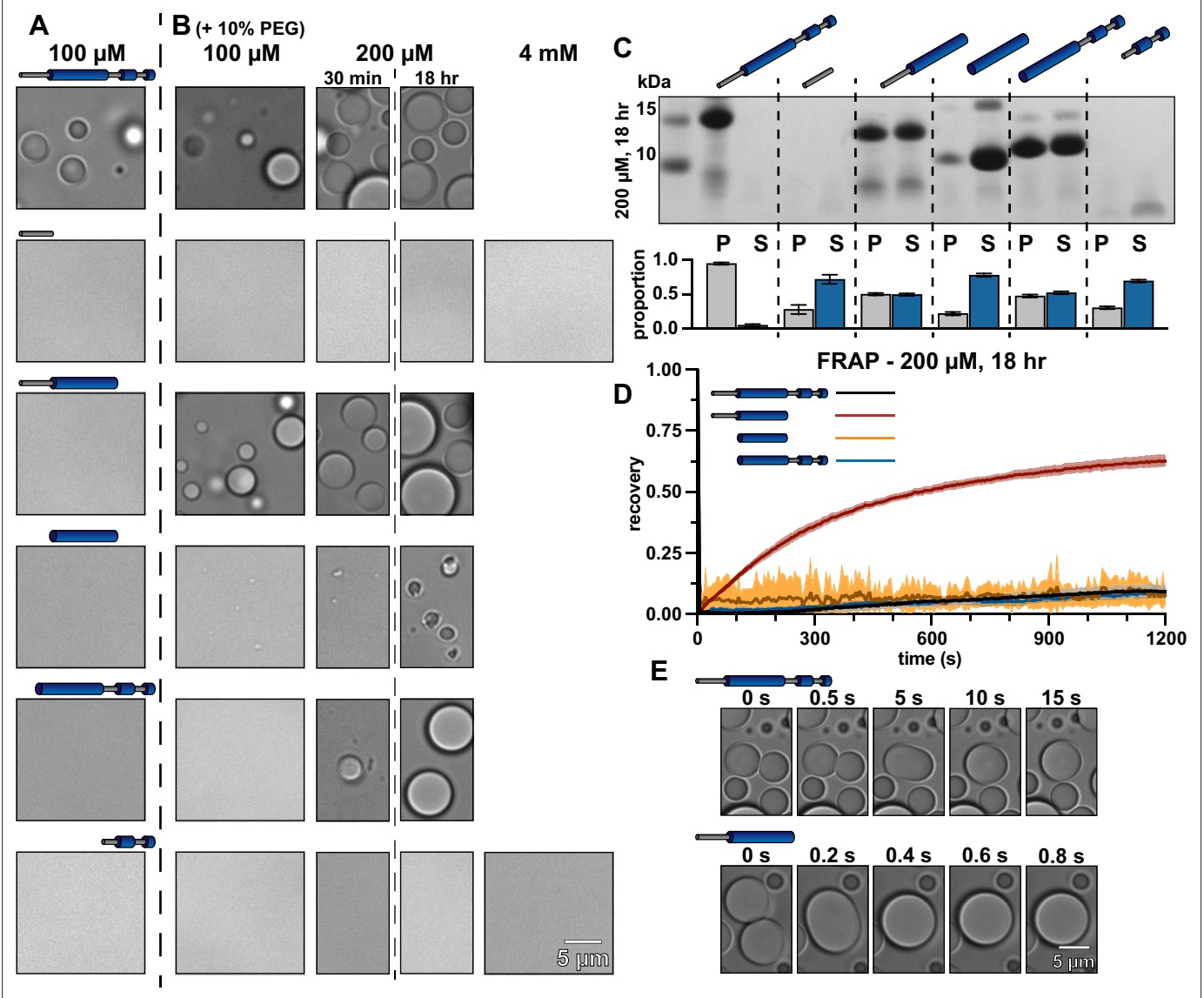

**Figure 4.** Truncations provide insight into the mechanisms of McdB condensate formation and stabilization. (**A**) Representative DIC microscopy images of full-length and truncation mutants of McdB at 100 μM in 150 mM KCl and 20 mM HEPES, pH 7.2. (**B**) As in (**A**), but with increasing protein concentration as indicated and with the addition of 10% PEG-8000. Scale bar applies to all images. All domains are required for FL level condensate formation. (**C**) Condensates at 200 μM after 18 hr were pelleted (P) and run on a sodium dodecyl sulfate–polyacrylamide gel electrophoresis (SDS–PAGE) gel along with the associated supernatant (S) (*top*). P and S band intensities were then quantified (*bottom*). Mean and standard deviation (SD) from 3 replicates are shown. (**D**) Fluorescence recovery after photobleaching (FRAP) of condensates at the indicated condition reveal an increase in dynamics when the N-term intrinsically disordered region (IDR) is present without the C-terminal domain (CTD). Mean and SD from 7 replicates are shown (**E**) Condensates containing the N-term IDR fuse orders of magnitude more quickly in the absence of the CTD, suggesting a stabilizing interaction between the two termini.

The online version of this article includes the following source data and figure supplement(s) for figure 4:

**Source data 1.** Sodium dodecyl sulfate–polyacrylamide gel electrophoresis (SDS–PAGE) gel corresponding to the pelleting assay described in *Figure 4C*.

**Source data 2.** Spreadsheet containing the raw data for gel quantification graphs shown in *Figure 4C*.

**Source data 3.** Spreadsheet containing the raw data for fluorescence recovery after photobleaching (FRAP) curves shown in *Figure 4D*.

**Figure supplement 1.** McdB forms liquid-like condensates in both Ficoll and polyethylene glycol.

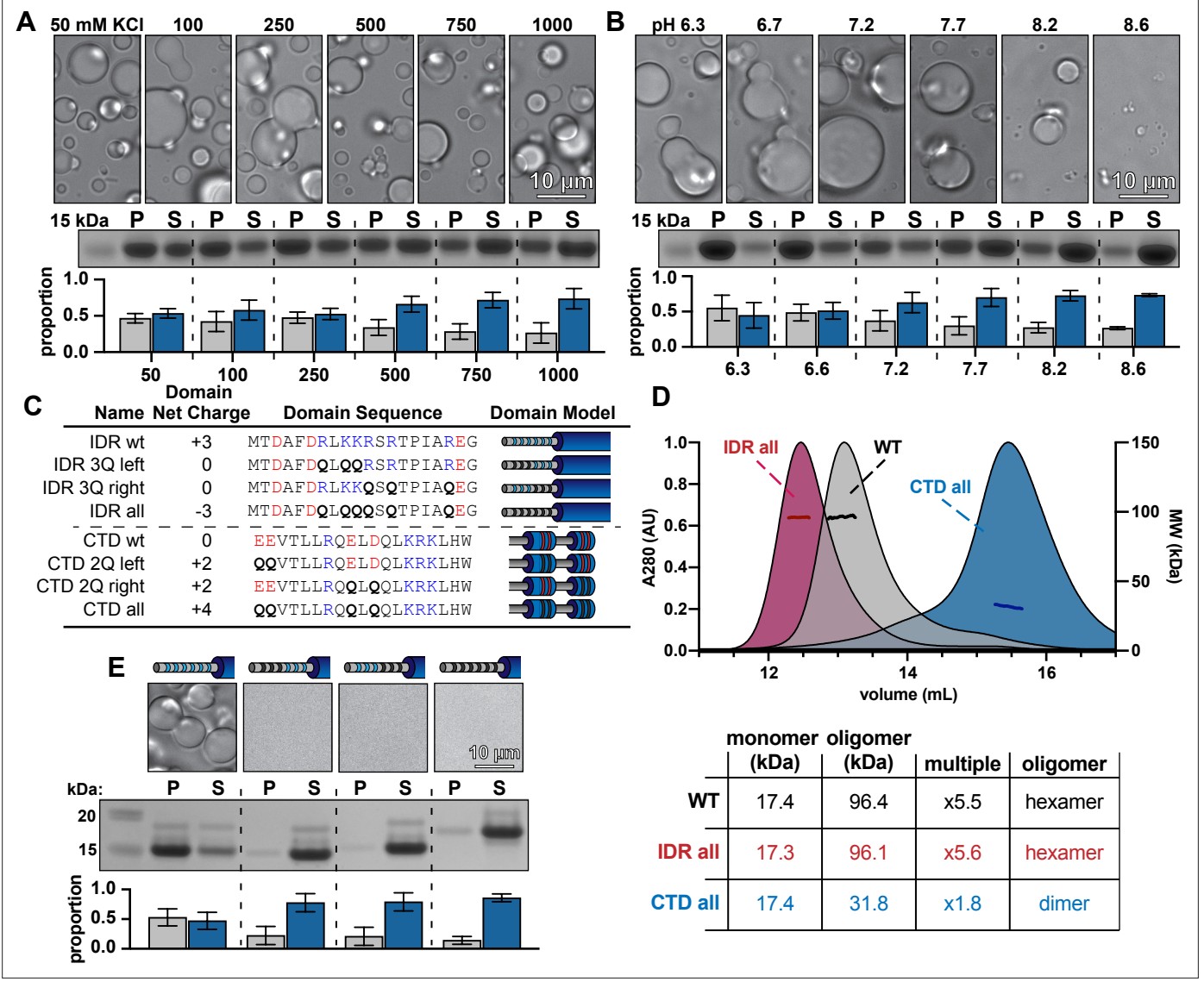

**Figure 5.** McdB condensates can be solubilized by mutating basic residues in the N-terminal intrinsically disordered region (IDR) without affecting McdB structure. (**A**) Representative DIC microscopy images of 50 µM McdB in 20 mM HEPES, pH 7.2 and increasing KCl concentration (*top*). Scale bar applies to all images. McdB condensates were pelleted (P) and run on a sodium dodecyl sulfate–polyacrylamide gel electrophoresis (SDS–PAGE) gel along with the associated supernatant (S) (*middle*). P and S band intensities were then quantified (*bottom*). Mean and standard deviation (SD) from 3 replicates are shown. (**B**) As in (**A**), except salt was held constant at 100 mM KCl and the pH was increased as indicated. (**A**) and (**B**) implicate stabilizing basic residues (see *Figure 5—figure supplement 2*). (**C**) Table showing the net charge and amino acid sequence of wild-type McdB compared to the glutamine (Q)-substitution mutants in both the N-term IDR and C-terminal domain (CTD). Acidic and basic residues in the IDR are colored red and blue, respectively. Q-substitutions are bolded. Graphical models of the McdB variants are also provided. (**D**) Size-exclusion chromatography coupled to multi-angle light scattering (SEC-MALS) of WT McdB compared to the full Q-substitution mutants from both the N- and C-termini. (*Below*) Table summarizing the SEC-MALS data, showing that mutations to the IDR does not affect oligomerization, while mutations to the CTD destabilize the trimer-of-dimers hexamer (see *Figure 5—figure supplement 3*). (**E**) Representative DIC microscopy images for WT and IDR Q-substitution mutants of McdB at 100 µM in 150 mM KCl and 20 mM HEPES, pH 7.2 (*top*). Scale bar applies to all images. McdB condensates were pelleted (P) and run on a sodium dodecyl sulfate–polyacrylamide gel electrophoresis (SDS–PAGE) gel along with the associated supernatant (S) (*middle*). P and S band intensities were then quantified (*bottom*). Mean and standard deviation (SD) of 3 replicates are shown.

The online version of this article includes the following source data and figure supplement(s) for figure 5:

**Source data 1.** Sodium dodecyl sulfate–polyacrylamide gel electrophoresis (SDS–PAGE) gel corresponding to the pelleting assay described in *Figure 5A*.

**Source data 2.** Sodium dodecyl sulfate–polyacrylamide gel electrophoresis (SDS–PAGE) gel corresponding to the pelleting assay described in *Figure 5B*.

*Figure 5 continued on next page*

*Figure 5 continued*

**Source data 3.** Sodium dodecyl sulfate–polyacrylamide gel electrophoresis (SDS–PAGE) gel corresponding to the pelleting assay described in *Figure 5E*.

**Source data 4.** Spreadsheet containing the raw data for gel quantification graphs shown in *Figure 5A, B*.

**Source data 5.** Spreadsheet containing the raw data for size-exclusion chromatography coupled to multi-angle light scattering (SEC-MALS) curves shown in *Figure 5D* and an editable version of the associated table.

**Source data 6.** Spreadsheet containing the raw data for gel quantification graphs shown in *Figure 5E*.

**Figure supplement 1.** Multidimensional phase diagrams for *Se7942* McdB.

**Figure supplement 1—source data 1.** Spreadsheet containing the raw data for turbidity assay graphs shown in *Figure 5—figure supplement 1S*.

**Figure supplement 2.** Alanine scanning of basic residues in the N- and C-termini of McdB.

**Figure supplement 3.** Mutations to the C-terminal domain (CTD) destabilize the trimer-of-dimers hexamer.

**Figure supplement 3—source data 1.** Spreadsheet containing the raw data for size-exclusion chromatography coupled to multi-angle light scattering (SEC-MALS) curves shown in *Figure 5—figure supplement 3S* and an editable version of the associated table.

Together with our previous data showing that both the IDR and CTD modulate condensation, we focused on the basic residues within these two domains. By making a series of alanine substitutions (*Figure 5—figure supplement 2A*), we found that removing positive charge in the IDR, but not the CTD, caused a loss of condensates (*Figure 5—figure supplement 2B*). However, we also found that substituting charged residues for a more hydrophobic residue like alanine caused protein aggregation (*Figure 5—figure supplement 2B*) as found for other proteins (*Xue et al., 2019*). Therefore, going forward, we transitioned to substituting these charged residues with polar glutamines (*Figure 5C*), to specifically affect charge and not hydrophilicity.

Data from this report and our previous study (*MacCready et al., 2020*) suggest a potential electrostatic interaction between the positively charged IDR and negatively charged residues of the CTD. We therefore created another series of substitutions where we changed either positive charge in the IDR or negative charge in the CTD to glutamines. Before assessing condensate formation, we first performed SEC-MALS to verify these mutations had no major impact on McdB structure. Substituting all six basic residues to glutamines in the IDR had no effect on McdB hexamerization (*Figure 5D*). On the other hand, only two substitutions in the CTD were enough to partially destabilize the hexamer (*Figure 5—figure supplement 3*), and four substitutions produced mainly McdB dimers (*Figure 5D*). As a result, we were unable to parse out the different roles of the CTD in McdB oligomerization versus potential interactions with the IDR involved in condensate formation. Importantly, however, we determined that removing only three positively charged residues in the IDR solubilized McdB condensates (*Figure 5E*) without affecting protein structure (*Figure 6—figure supplement 1*) or hexamerization (*Figure 5D*).

## McdB condensate formation is tunable through changes in IDR net charge

Substituting only three basic residues in the IDR (net charge −3) completely solubilized McdB condensates under our standard conditions (*Figure 5E*). We set out to determine the effect of fewer mutations in the IDR on condensate solubility. Pairs of basic residues in the IDR were substituted with glutamines, leaving an IDR net charge of +1 (*Figure 6A*). All +1 mutants still formed condensates, albeit smaller and fewer than that of wild-type McdB (*Figure 6B*).

The data suggest McdB condensate formation is tunable though changes to the net charge of the IDR. If correct, the triplet substitution mutants (IDR net charge 0) may still be capable of forming condensates at higher protein concentrations. Indeed, when we added a crowding agent, the net charge 0 mutants formed condensates (*Figure 6C*). Moreover, a gradual increase in the proportion of McdB in the soluble phase was revealed as we incrementally removed positive charge from the IDR. Removing all six positive residues (net charge +3) still completely solubilized McdB even in the presence of a crowder (*Figure 6C*). Importantly, McdB mutants with the same IDR net charge, but with different residues substituted, showed similar changes to condensate solubility (*Figure 6C*). Substitution position showed slight differences in condensate size, but the overall effect on solubility was the

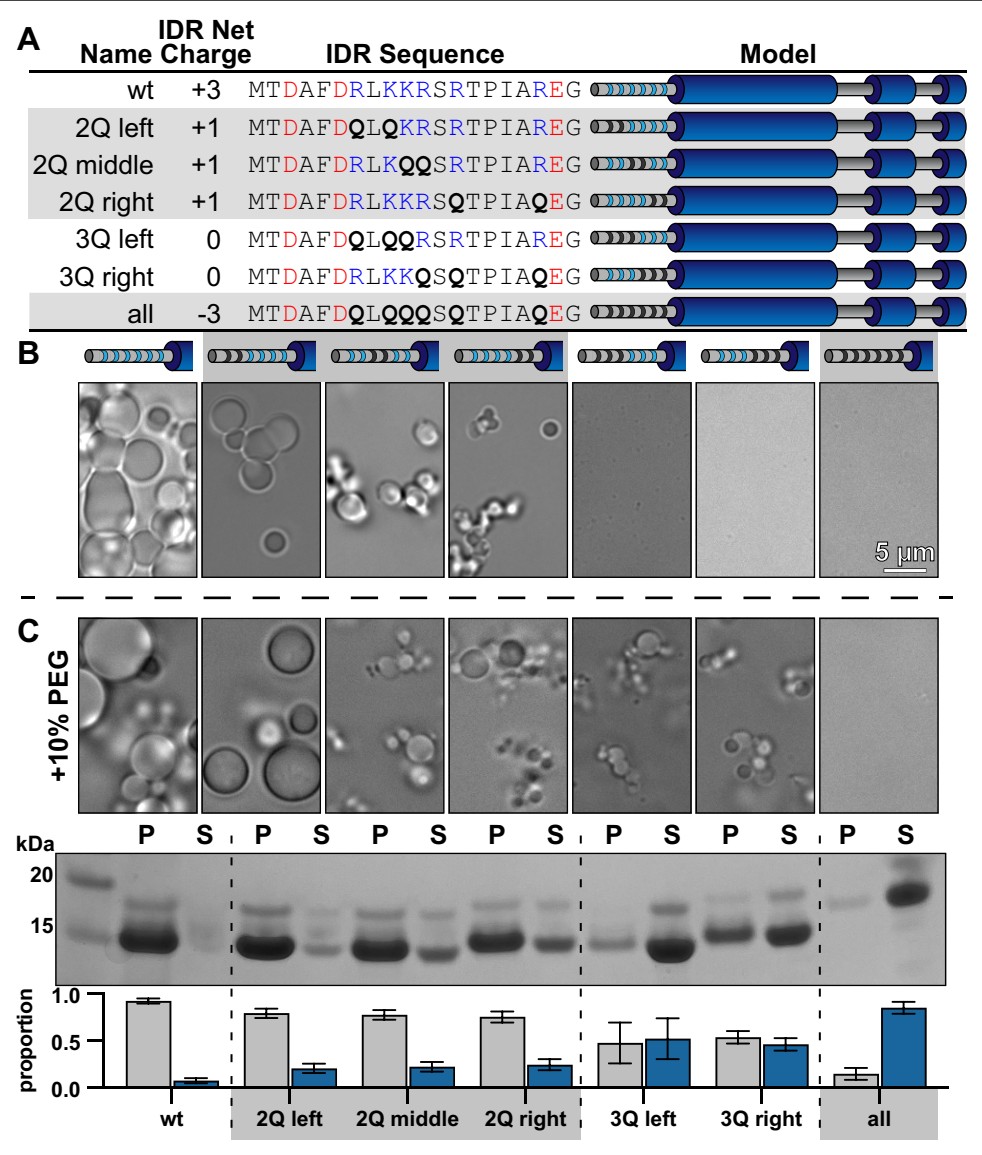

**Figure 6.** Net charge of the intrinsically disordered region (IDR) can be used to tune the solubility of McdB condensates. (**A**) Table showing the net charge and N-terminal IDR sequence of wild-type McdB compared to the glutamine (Q)-substitution mutants. Acidic and basic residues in the IDR are colored red and blue, respectively. Q-substitutions are bolded. Graphical models of the McdB variants are also provided where blue stripes represent the six basic residues in the IDR. Black stripes represent the location of the Q-substitutions. (**B**) Representative DIC microscopy images for wild-type and the Q-substitution mutants of McdB at 100 µM in 150 mM KCl and 20 mM HEPES, pH 7.2. Scale bar applies to all images. (**C**) As in (**B**), but with the addition of 10% PEG-8000 (*top*). McdB condensates were pelleted (P) and run on a sodium dodecyl sulfate–polyacrylamide gel electrophoresis (SDS–PAGE) gel along with the associated supernatant (S) (*middle*). P and S band intensities were then quantified (*bottom*). Mean and standard deviation (SD) from 3 replicates are shown.

The online version of this article includes the following source data and figure supplement(s) for figure 6:

**Source data 1.** Sodium dodecyl sulfate–polyacrylamide gel electrophoresis (SDS–PAGE) gel corresponding to the pelleting assay described in *Figure 6C*.

**Source data 2.** Spreadsheet containing the raw data for gel quantification graphs shown in *Figure 6C*.

**Figure supplement 1.** Circular dichroism (CD) spectra of wild-type McdB and N-terminal glutamine-substitution mutants.

**Figure supplement 1—source data 1.** Spreadsheet containing the raw data for circular dichroism (CD) curves shown in *Figure 6—figure supplement 1S*.

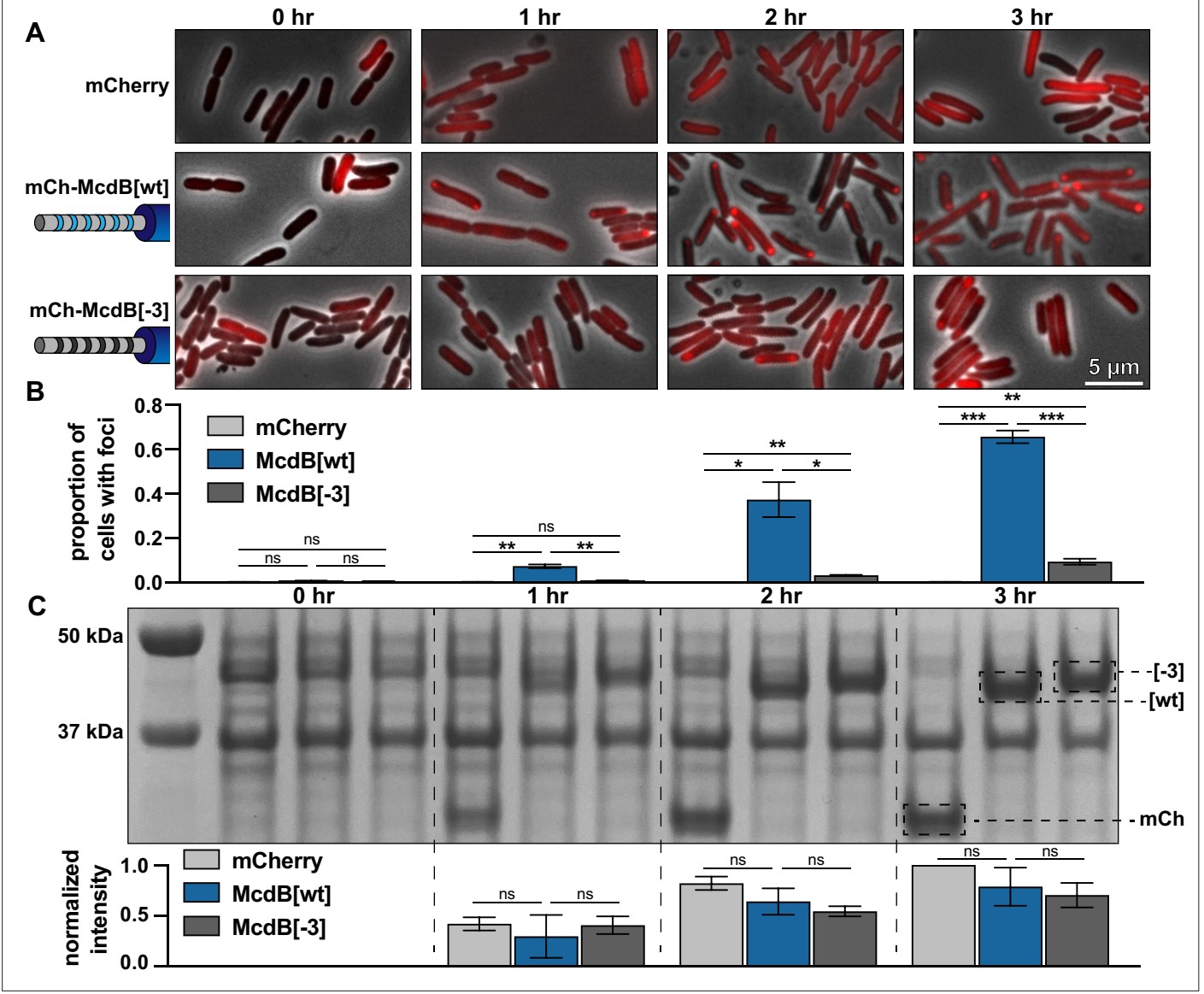

**Figure 7.** Net charge of the intrinsically disordered region (IDR) affects McdB solubility in *E. coli*. (**A**) Representative fluorescence microscopy images monitoring the expression of the indicated constructs over time. Scale bar applies to all images. (**B**) Quantification of the proportion of cells containing foci from the images represented in (**A**). All quantifications were done on >300 cells and from *n* = 3 technical replicates. Reported values represent means with standard deviation (SD). *p < 0.05, **p < 0.01, ***p < 0.001 by Welch's *t* test. (**C**) Sodium dodecyl sulfate–polyacrylamide gel electrophoresis (SDS–PAGE) of cell lysates from the time course represented in (**A**). All samples were standardized to the same $OD_{600}$ prior to loading. The expected MWs of the three constructs indicated are: mCherry 26.8 kDa; mCh-McdB[wt] 44.6 kDa; mCh-McdB[−3] 44.5 kDa. Normalized intensities from the indicated bands were quantified from 3 biological replicates (*below*). Reported values represent means with SD. Data were analyzed via Welch's *t* test.

The online version of this article includes the following source data for figure 7:

**Source data 1.** Sodium dodecyl sulfate–polyacrylamide gel electrophoresis (SDS–PAGE) gel corresponding to the protein expression levels described in *Figure 7C*.

**Source data 2.** Spreadsheet containing the raw data for foci count quantification graphs shown in *Figure 7B*.

**Source data 3.** Spreadsheet containing the raw data for gel quantification graphs shown in *Figure 7C*.

same within each charge grouping. Together, the data show that it is the net charge of the IDR, and not a specific basic residue, that is critical for mediating condensate solubility.

## Net charge of the IDR affects McdB condensation in *E. coli*

To determine if the IDR can be used to tune McdB solubility in cells, we induced expression of fluorescent fusions of mCherry with both wild-type McdB (McdB[wt]) and the full glutamine-substitution mutant, with an IDR net charge of −3 ('McdB[−3]') in *E. coli* MG1655. As protein concentration increased, McdB[wt] formed polar foci that coexisted with a dilute cytoplasmic phase (*Figure 7A*), similar to the dense and dilute phases of McdB in vitro. After 3 hr of expression, nearly 70% of cells with McdB[wt] adopted this two-state regime (*Figure 7B*). The foci were indeed driven by McdB, as mCherry alone remained diffuse (*Figure 7A, B*). McdB[−3], on the other hand, was considerably more soluble than wild-type, where even after 3 hr of expression <10% of cells contained foci (*Figure 7A, B*). The change in solubility was not due to differences in protein levels or due to cleavage of the fluorescent tag (*Figure 7C*), but instead represents an increased solubility due to the IDR substitutions. Together the data show that adjustments to the net charge of the IDR can also affect McdB condensate solubility in vivo.

## McdB[−3] causes mispositioned carboxysomes, likely due to an inability to interact with McdA

Having identified a mutant that solubilizes McdB condensates, without affecting structure or hexamerization, we set out to determine its influence on carboxysome positioning in *Se7942*. mNeonGreen (mNG) was N-terminally fused to either McdB[wt] or McdB[−3] and expressed at its native locus. The small subunit of Rubisco (RbcS) was C-terminally fused to mTurquoise (mTQ) to image carboxysomes. As shown previously (*MacCready et al., 2018*), mNG-McdB[wt] supported well-distributed carboxysomes along the cell length (*Figure 8A*, *Figure 8—figure supplement 1*). The mNG-McdB[−3] strain, on the other hand, displayed carboxysome aggregates. However, it is important to note that McdA is a ParA/MinD family ATPase, which typically interact with their adaptor proteins via basic resides in the N-terminus of the adaptor protein, analogous to McdB (*Radnedge et al., 1998*; *Ravin et al., 2003*; *Barillà et al., 2007*; *Ah-Seng et al., 2009*; *Ghasriani et al., 2010*; *Schumacher et al., 2021*). Therefore, it is highly likely that one or more of the basic residues removed from McdB[−3] not only modulate condensate formation, but also mediate McdA interaction. A loss in McdA interaction would explain the carboxysome aggregation phenotype, as we have shown previously (*MacCready et al., 2018*; *Funnell, 2016*). To investigate this possibility, we knocked out McdA in the McdB[−3] mutant and found no significant differences in carboxysome mispositioning compared to the *ΔmcdA* strain alone (*Figure 8—figure supplement 1*). Together, the data suggest that carboxysome aggregation in the McdB[−3] strain is due to this mutant's inability to interact with McdA, and is not necessarily due to the effects these mutations have on McdB condensate formation.

## Condensate-defective McdB[−3] has a reduced cytoplasmic phase and associates with carboxysomes with lowered Rubisco content

Although carboxysome mispositioning by McdB[−3] cannot be directly ascribed to defects in McdB condensate formation specifically, two other observed phenotypes are not explained by a loss in McdA interaction. First, McdB[−3] still strongly colocalized with carboxysomes (Pearson's correlation coefficient [PCC] = 0.86 ± 0.01, $n > 10,000$ cells), similar to that of both McdB[wt] (PCC = 0.81 ± 0.01, $n > 10,000$ cells), and association did not change in the absence of McdA (PCC = 0.86 ± 0.02, $n > 10,000$ cells) (*Figure 8A*). The data strongly suggest that condensate formation is not required for McdB to associate with carboxysomes.

Strikingly, however, the cytoplasmic phase observed for McdB[wt] was significantly lower for McdB[−3], both in the presence and absence of McdA (*Figure 8A*). Indeed, when quantifying the intensity ratio of carboxysome-associated McdB to that of the whole cell, McdB[−3] showed a significant deviation in the ratio that was independent of McdA (*Figure 8B*). The data show that while McdB condensate formation is not required for carboxysome association, without this activity, the cytoplasmic fraction of McdB notably declines.

Finally, we set out to directly determine the effect of McdB[−3] on the carboxysome itself by quantifying encapsulated Rubisco. Intriguingly, the McdB[−3] strains, with or without McdA,

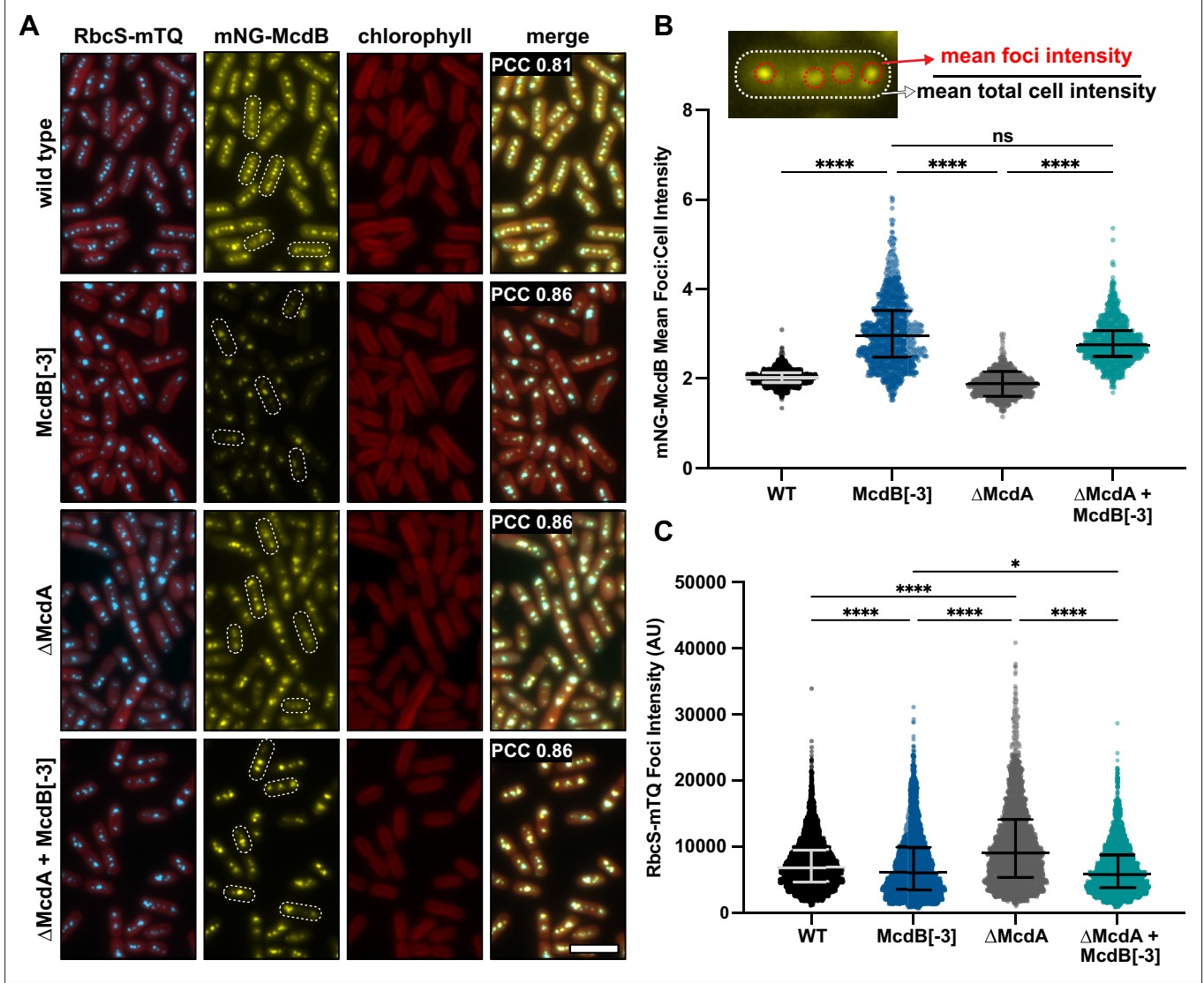

**Figure 8.** McdB[−3], which results in a high degree of condensate solubilization in vitro and in *E. coli*, alters the soluble fraction of McdB and carboxysome Rubisco levels in vivo. (**A**) Representative fluorescence microscopy images of the indicated strains. Scale bar = 5 μm and applies to all images. Pearson's correlation coefficients (PCCs) are shown for mNG-McdB and RbcS-mTQ for each strain. PCC values are means from >10,000 cells over 10 fields of view. (**B**) Quantification of (mean foci intensity/mean total cell intensities) for mNG-McdB of $n > 500$ cells. Medians and interquartile ranges are displayed. ****p < 0.001 based on Kruskal–Wallis analysis of variance (ANOVA). (**C**) Quantification of mean RbcS-mTQ foci intensity for $n > 500$ cells. Medians and interquartile ranges are displayed. *p < 0.05; ****p < 0.001 based on Kruskal–Wallis ANOVA.

The online version of this article includes the following source data and figure supplement(s) for figure 8:

**Source data 1.** Spreadsheet containing the raw data for graphs shown in *Figure 8B, C*.

**Figure supplement 1.** Deletion of McdA causes no additional loss of carboxysome positioning in McdB[−3] strain.

**Figure supplement 1—source data 1.** Spreadsheet containing the raw data for graphs shown in *Figure 8—figure supplement 1A, B*.

had carboxysomes with significantly lower Rubisco content as quantified by RbcS-mTQ intensity (*Figure 8C*). This finding was particularly striking in the *ΔmcdA* background because, as we have shown previously, deletion of McdA results in increased RbcS-mTQ foci intensity due to carboxysome aggregation (*Rillema et al., 2021*). But with McdB[−3], RbcS-mTQ intensity decreased, even with McdA deleted (*Figure 8C*). Together, these data show that McdB[−3] increases the carboxysome-bound to soluble McdB ratio and decreases Rubisco content in carboxysomes. These phenotypes are not explained by the loss of interaction with McdA, and are therefore potentially linked to defects in McdB condensate formation.

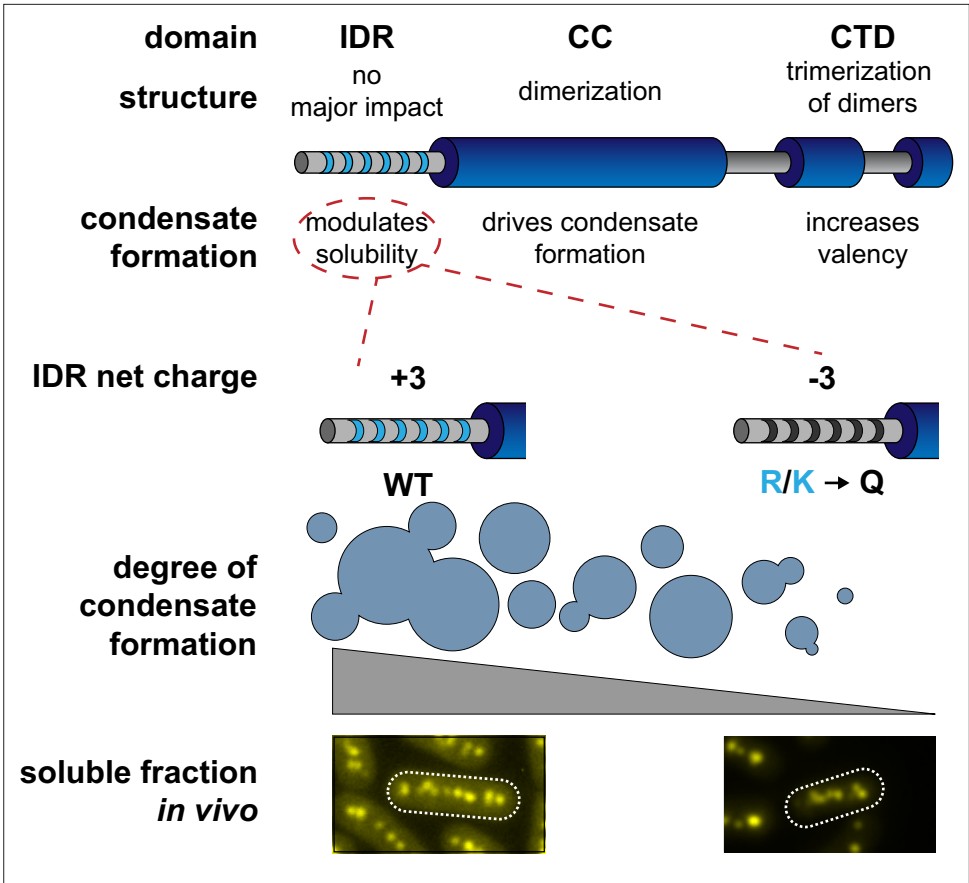

**Figure 9.** Proposed model of *Se7942* McdB domain structure and self-association. The central coiled-coil (CC) domain is necessary and sufficient for dimerization and driving condensate formation. The α-helical C-terminal domain (CTD) trimerize McdB dimers and increases the degree of condensate formation compared to the CC alone. The N-terminal intrinsically disordered region (IDR) does not affect oligomerization and increases the degree of condensate formation compared to the CC alone. Substituting basic residues (K/R) in the IDR to glutamines (Q) can tune condensate solubility in vitro without affecting McdB oligomerization. These mutations allowed us to identify in vivo phenotypes correlated specifically to McdB phase separation, including the relative amount of soluble McdB.

## Discussion

In this report, we generate an initial structural model of *Se7942* McdB based on several empirical and predictive approaches. We define a tripartite domain architecture with an N-terminal IDR, a stable CC domain, and a CTD consisting of several α-helices (*Figure 1*). We show that the CC dimerizes McdB and the CTD trimerizes the dimer, resulting in a trimer-of-dimers hexamer. The IDR had no impact on oligomerization (*Figure 2*). Next, we found that McdB forms condensates via pH-dependent PSCP, where condensates show time-dependent viscoelastic properties and McdB forms pH-dependent clusters at concentrations far below the observed $c_{sat}$ (*Figure 3*). Using truncations, we found that the CC domain drives condensate formation, the IDR modulates solubility, and the CTD provides further valency. Therefore, all three domains are required for achieving wild-type levels of condensate formation (*Figure 4*). We then identified positive residues in McdB important for stabilizing condensates. By performing scanning mutagenesis in both the IDR and CTD, we show that while mutations to the CTD destabilize the McdB hexamer, substituting out basic residues in the IDR solubilized condensates without affecting McdB structure or oligomerization (*Figure 5*). These findings allowed us to design a series of mutants where the net charge of the IDR tuned McdB condensate solubility both in vitro (*Figure 6*) and in vivo (*Figure 7*). Lastly, we found that a solubilized McdB mutant, McdB[−3], impacts the carboxysome-bound to soluble McdB ratio in the cell, as well as Rubisco content in carboxysomes (*Figure 8*). Overall, we determined McdB domain architecture, its oligomerization domains, regions

required for condensate formation, and how to fine-tune condensate solubility, allowing us to link McdB condensate formation to potential functions in vivo (*Figure 9*).

## McdB condensate formation follows a nuanced, multidomain mechanism

As the field of biomolecular condensates advances, more nuanced mechanisms are arising that describe combinations of condensate-driver domains, solubility modulators, and influences of oligomerization. We see here that different domains of McdB influence condensate formation via effects on solubility, self-association, and network formation. For many proteins, IDRs have been shown to be necessary and sufficient for driving condensate formation to a degree that is comparable to the full-length protein (*Elbaum-Garfinkle et al., 2015*; *Muiznieks et al., 2018*; *Kim et al., 2013*; *Darling et al., 2018*; *Shapiro et al., 2021*). The IDR of McdB, on the other hand, did not form condensates even at high protein concentrations and in the presence of a crowder. Furthermore, while deleting the IDR did not prevent McdB condensate formation, substituting only six basic residues in the IDR with glutamines solubilized condensates both in vitro and in *E. coli*. In line with our findings, recent models have described how charged residues within IDRs can serve as key modulators of protein phase separation via mediating interactions with the solvent (*Fossat et al., 2021b*; *Zeng et al., 2022*; *Fossat et al., 2021a*).

Glutamine-rich regions are known to be involved in stable protein-protein interactions such as in CCs and amyloids (*Fiumara et al., 2010*; *Michelitsch and Weissman, 2000*), and expansion of glutamine-rich regions in some condensate-forming proteins leads to amylogenesis and disease (*Kokona et al., 2014*; *Kwon et al., 2018*). Thus, one potential caveat here is that introduction of glutamines may lead to amylogenesis for McdB. However, when we introduced glutamines into the IDR of McdB, solubility was increased both in vitro and in vivo without any impact on hexamerization. These findings are especially striking for a condensate-forming protein as glutamines are generally thought to stabilize and coarsen protein condensates (*Wang et al., 2018a*). On the contrary, we see that increased glutamine content can solubilize condensates and is therefore largely context dependent. These findings expand our understanding of the molecular grammar of biomolecular condensates.

Oligomerization has also been found to influence protein condensate formation. For example, some proteins require oligomerization to provide the multivalency needed to form condensates (*Guillén-Boixet et al., 2020*; *Wang et al., 2018b*; *Marzahn et al., 2016*), where some IDRs only induce condensate formation when fused to an oligomerizing domain (*Bracha et al., 2018*). In some cases, oligomerization domains, like CCs, drive condensate formation and are modulated by other domains (*Ramirez et al., 2023*), similar to our findings here with McdB. Such is the case for the bacterial protein PopZ, where an oligomerization domain forms condensates with solubility modulated by an IDR (*Peeples and Rosen, 2021*).

Truncated proteins have been useful in the study of biomolecular condensates. But it is important to note that using truncation data alone to dissect modes of condensate formation can lead to erroneous models since entire regions of the protein are missing. However, data from our truncation and substitution mutants were entirely congruent. For example, deletion of the CTD or substitutions to this region caused destabilization of the hexamer to a dimer, and deletion of the IDR or substitutions to this region caused solubilization of condensates without affecting hexamer formation. Furthermore, it should be noted that the McdB constructs used in our in vitro assays were free from fluorescent proteins, organic dyes, or other modifications that may influence phase separation. Therefore, the observed material properties of these condensates have full dependence on the McdB sequence.

## McdB homologs have polyampholytic properties between their N- and C-termini

In our previous bioinformatic study, we found that McdB homologs possess features that enable condensate formation, including intrinsic disorder, low hydrophobicity, biased amino acid compositions, and multivalency (*MacCready et al., 2020*). Intriguingly, we also found that McdB proteins were potentially polyampholytes, with biphasic charge distributions between the NTD and CTD flanking the CC domain. For *Se7942* McdB, the N-terminal IDR has a pI of 10.8 and the CTD has a pI of 6.8, suggesting a potential electrostatic interaction between the two termini. The reason for such a shared feature was not obvious, but we proposed this polyampholytic nature was important for

McdB self-association. A structure of the CC domain of a plasmid-encoded McdB-like protein from the cyanobacterium *Cyanothece* sp. PCC 7424 displayed an antiparallel association to form a dimer (*Schumacher et al., 2019*). Antiparallel dimerization of the CC domains of *Se7942* McdB would align these oppositely charged extensions. Consistently, our truncation data provide evidence suggesting a condensate-stabilizing interaction between the N-terminal IDR and the CTD. Condensates from the IDR + CC construct, which lack the CTD, were highly dynamic. But with the CTD present, condensates fused slowly. We were unable to dissect how the CTD contributes to condensate formation as substitutions to the CTD caused hexamer destabilization. However, these results have set the stage for several future studies that will (1) probe the orientation of McdB subunits within the hexamer, (2) determine how basic residues in the IDR influence McdB condensate formation at a molecular level, and (3) identify residues in the CTD that may influence condensate formation.

## Considerations for McdB condensate formation in vivo

Investigating biomolecular condensates is particularly challenging in bacteria, largely because of resolution limits that require a combination of techniques, such as super-resolution microscopy (*Azaldegui et al., 2021*). These methods will certainly advance our understanding of McdB function in vivo. To this end, we began with biochemical approaches to identify mutation sets that affect condensate solubility in vitro and investigated the effects of these mutations in vivo. By mutating positively charged residues in the IDR of McdB to glutamines, we were able to solubilize condensates and shift McdB molecules into the dilute phase both in vitro and in *E. coli* cells. Surprisingly, this same mutation set seemed to have an opposite effect in *S. elongatus* cells, where the dilute phase was diminished and carboxysome-associated McdB foci became brighter (see *Figure 8*). The findings strongly suggest that McdB condensate formation is not required for foci formation in vivo when colocalizing with carboxysomes. Instead, our data suggest that phase separation may negatively regulate carboxysome binding. In vitro, condensate formation represents a major form of McdB self-association, sequestering away a large portion of McdB molecules into self-enriched condensates where they have a lessened interaction with the surrounding solvent. In vivo, these self-associations may similarly serve to regulate the amount of McdB free to interact with their carboxysome binding sites. For instance, if the positively charged N-terminal IDR of McdB interacts with the negatively charged CTD, and the CTD is also required for carboxysome binding, this association may interfere with carboxysome binding. Thus, by greatly reducing the tendency of McdB to self-associate, we may be increasing the amount of McdB free to interact with its carboxysome binding sites. We find such a model attractive as it provide a means for McdB to be pH regulated; an important factor in carboxysome biology as described below. To test this model, going forward, we will need to determine the region and residues of McdB required for carboxysome association and determine how this association relates to McdB condensate formation and its polyampholytic properties.

## pH as a potential regulator for McdB condensate solubility and its association with carboxysomes

In carbon-fixing organisms, the collection of processes that contribute to efficient carbon fixation are referred to as the carbon concentrating mechanism (CCM). The development of a model for the cyanobacterial CCM has provided insight into how different features, such the presence of carboxysomes, affect overall carbon capture (*Mangan and Brenner, 2014*). It was shown that incorporation of a pH flux into CCM models provides values more consistent with experimentation (*Mangan et al., 2016*). This updated 'pH-aware' model suggests that the carboxysome lumen maintains a lower pH than the cytoplasm via Rubisco proton production; with the cytoplasm being ~pH 8.5 and the carboxysome lumen being ~pH 7.5 (*Mangan et al., 2016*; *Long et al., 2021*). Here, we show pH as a major regulator of McdB condensate solubility in vitro. Furthermore, we report a potential link between McdB condensate solubility and regulating both the carboxysome-bound fractions of McdB as well as Rubisco content within carboxysomes. We speculate that intracellular differences in pH may influence McdB self-association and thus regulate carboxysome binding as described in our model in the previous section.

Future studies will determine the nature of McdB association with carboxysomes, and how condensate formation influences this association. For example, we have shown that McdB strongly associates with carboxysome shell proteins via bacterial two-hybrid assays (*MacCready et al., 2018*). It

is attractive to speculate that McdB undergoes pre-wetting interactions with the 2D surface of the carboxysome shell, which then nucleates surface-assisted condensation. Such 2D interactions would significantly impact McdB phase boundaries (*Ditlev, 2021*). It is also possible that McdB phase separation directly modulates carboxysome fluidity and Rubisco content. Carboxysomes were traditionally thought of as paracrystalline, but recent data in *S. elongatus* show that carboxysome biogenesis begins with Rubisco forming a condensate with its linker protein CcmM (*Wang et al., 2019*). Intriguingly, our bacterial two-hybrid assays have shown that, in addition to shell proteins, McdB also interacts strongly with CcmM. It is therefore possible that McdB and its phase separation activity influences carboxysome composition, fluidity, and function through interactions with the Rubisco-CcmM condensate. Such a model helps explain defects in Rubisco loading in our condensation-deficient McdB stain. Teasing apart the stable protein–protein interactions between McdB and carboxysomes from the dynamic processes governing condensate solubility will therefore be of significant importance.

## Partner proteins of ParA/MinD ATPases form condensates with functional implications

McdA is a member of the ParA/MinD family of ATPases that position a wide array of cellular cargos, including plasmids, chromosomes, and the divisome (*Lutkenhaus, 2012*; *Vecchiarelli et al., 2012*; *Funnell, 2016*). A partner protein acts as the adaptor, linking the positioning ATPase to its respective cargo. McdB is the adaptor protein that links McdA to the carboxysome. Here, we show that McdB forms condensates both in vitro and in vivo, and the solubility of these condensates influences the nature of McdB association with carboxysomes. Intriguingly, condensate formation by other adaptor proteins has been proposed to play important roles in mediating interactions with other cargos positioned by ParA/MinD ATPases. For example, ParA ATPases are responsible for the spatial regulation of chromosomes and plasmids that are bound by the adaptor protein ParB (*Lutkenhaus, 2012*; *Vecchiarelli et al., 2012*; *Funnell, 2016*). The exact nature of the interaction between ParB and DNA remains a vibrant area of research (*Debaugny et al., 2018*), but recent reports show that ParB–DNA complexes behave as dynamic, liquid-like condensates both in vitro (*Babl et al., 2022*) and in vivo (*Guilhas et al., 2020*), implementing condensate formation as an underlying mechanism. Another example is the co-complex of adaptor proteins, PomX and PomY, for the ParA/MinD ATPase called PomZ. PomXYZ is responsible for spatially regulating division sites in some bacteria (*Schumacher et al., 2017*). A recent study has shown that PomY forms condensates that nucleate GTP-dependent FtsZ polymerization, suggesting a novel mechanism for positioning cell division (*Ramm et al., 2022*). It is intriguing to speculate why such disparate adaptor proteins could use condensate formation as an underlying mechanism in localizing to vastly different cargos. The ability to undergo changes in density at specific locations and times within the cell may confer an advantage to these highly dynamic spatial regulation systems.

## Tunable protein condensates as useful tools for synthetic biology

A useful property of biomolecular condensates is the ability to regulate enzyme activity (*O'Flynn and Mittag, 2021*). Specific chemistries within condensates can affect the degree to which certain metabolites and enzymes are soluble within the dense phase. Thus, condensates can serve as reaction centers that regulate the overall metabolism of a cell by transiently altering the activities of key reactions. For example, it has been shown that certain scaffolding proteins can form phase separated condensates with Rubisco (*Wang et al., 2019*; *Oltrogge et al., 2020*). It is speculated that these Rubisco condensates were the original CCM, which then led to the evolution of carboxysomes and the modern CCM (*Long et al., 2021*). An exciting future direction for the field of biomolecular condensates is the prospect of designing condensate forming scaffolding proteins that can recruit specific enzymes, such as Rubisco, and implementing these designer enzyme condensates in synthetic cells to engineer metabolism (*Peeples and Rosen, 2021*; *Lasker et al., 2021*; *O'Flynn and Mittag, 2021*).

Here, we find that the IDR of McdB, which does not itself drive condensate formation, is amenable to mutations that fine-tune McdB condensate properties. The bacterial protein PopZ has already been engineered to fine-tune condensate formation and has also been developed as a tool called the 'PopTag', which endows condensate forming activity to fusion proteins expressed in a variety of cell types, including human cells (*Lasker et al., 2021*). As McdB and PopZ differ at the primary, secondary, tertiary, and quaternary levels, it is expected that they could be used as separate tags to

design coexisting but immiscible condensates, thus expanding design potential and the repertoire of condensate-related tools for synthetic biology.

## Materials and methods

### Protein expression and purification

Wild-type and mutant variants of McdB were expressed with an N-terminal His-SUMO tag off a pET11b vector in *E. coli* BL21-AI (Invitrogen). All cells were grown in LB + carbenicillin (100 µg/ml) at 37°C unless otherwise stated. One liter cultures used for expression were inoculated using overnight cultures at a 1:100 dilution. Cultures were grown to an $OD_{600}$ of 0.5 and expression was induced using final concentrations of isopropyl ß-D-1-thiogalactopyranosideat (IPTG) 1 mM and L-arabinose at 0.2%. Cultures were grown for an additional 4 hr, pelleted, and stored at −80°C.

Pellets were resuspended in 30 ml lysis buffer [300 mM KCl; 50 mM Tris–HCl pH 8.4; 5 mM 2-Mercaptoethanol (BME); 50 mg lysozyme (Thermo Fisher); protease inhibitor tablet (Thermo Fisher)] and sonicated with cycles of 10 s on, 20 s off at 50% power for 7 min. Lysates were clarified via centrifugation at 15,000 rcf for 30 min. Clarified lysates were passed through a 0.45-µm filter and loaded onto a 1-ml HisTrap HP (Cytiva) equilibrated in buffer A [300 mM KCl; 50 mM Tris–HCl, pH 8.4; 5 mM BME]. Columns were washed with 5 column volumes of 5% buffer B [300 mM KCl; 20 mM Tris–HCl, pH 8.4; 5 mM BME; 500 mM imidazole]. Elution was performed using a 5–100% gradient of buffer B via an AKTA Pure system (Cytiva). Peak fractions were pooled and diluted with buffer A to a final imidazole concentration of <100 mM. Ulp1 protease was added to a final concentration of 1:100 protease:sample, and incubated overnight at 23°C with gentle rocking. The pH was then adjusted to ~10 and samples were concentrated to a volume of <5 ml, passed through a 0.45-µm filter and passed over a sizing column (HiLoad 16/600 Superdex 200 pg; Cytiva) equilibrated in buffer C [150 mM KCl; 20 mM CAPS, pH 10.2; 5 mM BME; 10% glycerol]. Peak fractions were pooled, concentrated, and stored at −80°C.

### Proteolysis and N-terminal sequencing

Proteolysis was performed on *Se7942* McdB at 30 µM in buffer containing 150 mM KCl, 50 mM HEPES, pH 7.7, and 2 mM BME. Trypsin protease (Thermo Fisher) was added at a 1:100 ratio of protease:protein. The reaction was incubated at 30°C and samples were quenched at the indicated time points by diluting into 4× Laemmli SDS–PAGE sample buffer containing 8% SDS. Degradation over time was visualized by running time points on a 4–12% Bis-Tris NuPAGE gel (Invitrogen) and staining with InstantBlue Coomassie Stain (Abcam).

Bands that were N-terminally sequenced were separated via SDS–PAGE as above, but transferred to a polyvinylidene difluoride (PVDF) membrane (Bio-Rad) prior to staining. Transfer of bands was performed using a Trans-Blot Turbo Transfer System (Bio-Rad). N-terminal sequences of these bands were then determined using Edman degradation.

### Circular dichroism

For all protein samples analyzed, far-UV CD spectra were obtained using a J-1500 CD spectrometer (Jasco). All measurements were taken with 250 µl of protein at 0.25 mg/ml in 20 mM KPi, pH 8.0. Measurements were taken using a quartz cell with a path length of 0.1 cm. The spectra were acquired from 260 to 190 nm with a 0.1-nm interval, 50 nm/min scan speed, and at 25°C unless otherwise stated.

### Microscopy of protein condensates

Samples for imaging were set up in 16-well CultureWells (Grace BioLabs). Wells were passivated by overnight incubation in 5% (wt/vol) Pluronic acid (Thermo Fisher), and washed thoroughly with the corresponding buffer prior to use. All condensate samples were incubated for 30 min prior to imaging unless otherwise stated. For experiments where samples were imaged across pH titrations, the following buffers were used: phosphate buffer for pH 6.3–6.7, HEPES for pH 7.2–7.7, and Tris–HCl for 8.2–8.6. Imaging of condensates was performed using a Nikon Ti2-E motorized inverted microscope (60× DIC objective and DIC analyzer cube) controlled by NIS Elements software with a Transmitted

LED Lamp house and a Photometrics Prime 95B Back-illuminated sCMOS Camera. Image analysis was performed using Fiji v 1.0.

## Fluorescence recovery after photobleaching

All FRAP measurements were performed using the indicated protein concentration with the addition of 1:1000 mNG-McdB based on molarity. All fluorescence imaging was performed using a Nikon Ti2-E motorized inverted microscope controlled by NIS Elements software with a SOLA 365 LED light source, a ×100 objective lens (Oil CFI60 Plan Apochromat Lambda Series for DIC), and a Photometrics Prime 95B Back-illuminated sCMOS camera. mNG signal was acquired using a 'GFP' filter set [excitation, 470/40 nm (450–490 nm); emission, 525/50 nm (500–550 nm); dichroic mirror, 495 nm]. Bleaching was conducted with a 405-nm laser at 40% power (20 mW) with a 200-µs dwell time. Recovery was monitored with a time-lapse video with 5-s intervals for 20 min. Image analysis was done in Fiji v 1.0. Intensities from bleached regions of interest were background subtracted and normalized using an unbleached condensate to account for any full field of view photobleaching. The values for each condensate were then normalized such that a value of 1 was set to the pre-bleach intensity and a value of 0 was set to the intensity immediately post-bleaching. Data were exported, further tabulated, graphed, and analyzed using GraphPad Prism 9.0.1 for macOS (GraphPad Software, San Diego, CA, https://www.graphpad.com).

## Dynamic light scattering

All sizing and polydispersity measurements were carried out on an Uncle by Unchained Labs (USA) at 25°C in triplicate. Samples were adjusted to the indicated concentrations in 150 mM KCl and 20 mM of the following buffers based on pH: HEPES, pH 7.2; Tris–HCl, pH 8.2; CAPS, pH 10.2. Samples were analyzed both before and after a centrifugation step at 20,000 rcf for 10 min as indicated. Data were exported, further tabulated, graphed, and analyzed using GraphPad Prism 9.0.1 for macOS (GraphPad Software, San Diego, CA, https://www.graphpad.com).

## Size-exclusion chromatography

SEC was performed on full-length and truncated McdB proteins using a Superdex 200 Increase 10/300 GL (Cytiva) column connected to an AKTA pure system (Cytiva). 500 µl of sample at 1.5 mg/ml protein was passed through the column at 0.4 ml/min in buffer [150 mM KCl; 20 mM Tris–HCl, pH 8.2] while monitoring absorbance at 220 nm.

## SEC coupled to multi-angled light scattering

For each sample analyzed, 50 µl at 1.5 mg/ml was passed over an SEC column (PROTEIN KW-804; Shodex) at a flow rate of 0.4 ml/min in buffer containing 150 mM KCl and 20 mM Tris–HCl, pH 8.2. Following SEC, the samples were analyzed using an A280 UV detector (AKTA pure; Cytiva), the DAWN HELEOS-II MALS detector with an internal QELs (Wyatt Technology), and the Optilab T-rEX refractive index detector (Wyatt Technology). The data were analyzed to calculate mass using ASTRA 6 software (Wyatt Technology). Bovine serum albumin was used as the standard for calibration.

## Phase diagrams

Data for phase diagrams were collected using an Infinite M200 PRO plate reader (Tecan). Samples were set up in 96-well glass-bottom plates (Thomas Scientific) and absorbance at 350 nm was measured as previously described (*Alberti et al., 2019*). Reported values are averages of triplicates with buffer blanks subtracted, and error bars representing standard deviations. Protein concentration, KCl concentration, and pH values varied as indicated, but for each pH value tested, 20 mM of the following buffers were used: phosphate buffer for pH 6.3–6.7, HEPES for pH 7.2–7.7, and Tris–HCl for pH 8.2–8.6.

## Quantification of phase separation via centrifugation

Centrifugation was used to quantify the degree to which McdB and its variants condensed under certain conditions, as previously described (*Alberti et al., 2019*). Briefly, 100 µl of sample was incubated at the conditions specified for 30 min, and then centrifuged at 16,000 rcf for 2 min. The supernatant was removed and the pellet resuspended in an equal volume of McdB solubilization buffer

[300 mM KCl, 20 mM CAPS, pH 10.2]; McdB does not condense at pH 10.2. Samples were then diluted into 4× Laemmli SDS–PAGE sample buffer. Pellet and supernatant fractions were visualized on a 4–12% Bis-Tris NuPAGE gel (Invitrogen) by staining with InstantBlue Coomassie Stain (Abcam) for 1 hr and then destaining in water for 14–16 hr. The intensities of the bands were quantified using Fiji v 1.0 and resultant data graphed using GraphPad Prism 9.0.1 for macOS (GraphPad Software, San Diego, CA, https://www.graphpad.com).

## Expression and visualization of mCherry fusions in *E. coli*

All constructs were expressed off a plasmid from a pTrc promoter in *E. coli* MG1655. Overnight cultures grown in LB + carbenicillin (100 µg/ml) were diluted at 1:100 into AB medium + carbenicillin (100 µg/ml) supplemented with (0.2% glycerol; 10 µg/ml thiamine; 0.2% casein; 25 µg/ml uracil). Cultures were grown at 37°C to an $OD_{600} = 0.3$ and induced with 1 mM IPTG. Following induction, cultures were grown at 37°C and samples taken at the indicated time points.

Cells used for imaging were prepared by spotting 3 µl of cells onto a 2% UltraPure agarose + AB medium pad on a Mantek dish. Images were taken using Nikon Ti2-E motorized inverted microscope controlled by NIS Elements software with a SOLA 365 LED light source, a ×100 Objective lens (Oil CFI Plan Apochromat DM Lambda Series for Phase Contrast), and a Hamamatsu Orca Flash 4.0 LT + sCMOS camera. mCherry signal was imaged using a 'TexasRed' filter set (C-FL Texas Red, Hard Coat, High Signal-to-Noise, Zero Shift, excitation: 560/40 nm [540–580 nm], emission: 630/75 nm [593–668 nm], dichroic mirror: 585 nm). Image analysis was performed using Fiji v 1.0.

To monitor expression levels, cells were harvested via centrifugation at the indicated time points, and resuspended in 4× Laemmli SDS–PAGE sample buffer to give a final $OD_{600} = 4$. Samples were boiled at 95°C and 10 µl were then run on a 4–12% Bis-Tris NuPAGE gel (Invitrogen). Bands were visualized by staining with InstantBlue Coomassie Stain (Abcam) for 1 hr and then destaining in water for 14–16 hr. Quantifying the normalized band intensities was performed using Fiji v 1.0.

## Growth and transformation of *S. elongatus* PCC 7942

All *S. elongatus* (ATCC 33912) strains were grown in BG-11 medium (Sigma) buffered with 1 g/l HEPES, pH 8.3. Cells were incubated with the following growth conditions: 60 µmol $m^{-2}$ $s^{-1}$ continuous LED 5600 K light, 32°C, 2% $CO_2$, and shaking at 130 RPM. Transformations of *S. elongatus* cells were performed as previously described (*Clerico et al., 2007*). Transformants were plated on BG-11 agar with 12.5 µg/ml kanamycin. Single colonies were picked and transferred liquid BG-11 medium with corresponding antibiotic concentrations. Complete gene insertions and absence of the wild-type gene were verified via PCR, and cultures were removed from antibiotic selection prior to imaging.

## Live cell fluorescence microscopy and analysis

100 µl of exponentially growing cells (OD750 ~ 0.7) were harvested and spun down at 4000 rcf for 1 min and resuspended in 10 µl fresh BG-11. 2 µl of the resuspension were then spotted on 1.5% UltraPure agarose (Invitrogen) + BG-11 pad on a 35-mm glass-bottom dish (MatTek Life Sciences). All fluorescence and phase-contrast imaging were performed using a Nikon Ti2-E motorized inverted microscope controlled by NIS Elements software with a SOLA 365 LED light source, a ×100 objective lens (Oil CFI Plan Apochromat DM Lambda Series for Phase Contrast), and a Photometrics Prime 95B back-illuminated sCMOS camera or Hamamatsu Orca-Flash 4.0 LTS camera. mNG-McdB variants were imaged using a 'YFP' filter set (C-FL YFP, Hard Coat, High Signal-to-Noise, Zero Shift, excitation: 500/20 nm [490–510 nm], emission: 535/30 nm [520–550 nm], dichroic mirror: 515 nm). RbcS-mTQ-labeled carboxysomes were imaged using a 'CFP' filter set (C-FL CFP, Hard Coat, High Signal-to-Noise, Zero Shift, excitation: 436/20 nm [426–446 nm], emission: 480/40 nm [460–500 nm], dichroic mirror: 455 nm). Chlorophyll was imaged using a 'TexasRed' filter set (C-FL Texas Red, Hard Coat, 583 High Signal-to-Noise, Zero Shift, excitation: 560/40 nm [540–580 nm], emission: 630/75 nm 584 [593–668 nm], dichroic mirror: 585 nm).

Image analysis including cell segmentation, quantification of foci number, intensities, and spacing were performed using Fiji plugin MicrobeJ 5.13n (*MacCready et al., 2020*; *Ducret et al., 2016*). Cell perimeter detection and segmentation were done using the rod-shaped descriptor with default threshold settings. Carboxysome foci detection was performed using the point function with tolerance of 700 and the sharpen image filter selected. McdB foci detection was performed using the smoothed

foci function with tolerance of 100, $Z$-score of 3, and the minimum image filter selected. PCCs were calculated using ImageJ plugin JaCoP (*Bolte and Cordelières, 2006*), and reported values represent means and standard deviations from >10,000 cells over 10 fields of view. Data were exported, further tabulated, graphed, and analyzed using GraphPad Prism 9.0.1 for macOS (GraphPad Software, San Diego, CA, https://www.graphpad.com).

## Acknowledgements

We would like to thank Dr. JK Nandakumar and Ritvija Agrawal for training and allowing us to use their SEC-MALS system. Dr. Henriette Remmer at the University of Michigan Proteomics & Peptide Synthesis Core for help with N-terminal sequencing. The National Crystallization Center at the Hauptman-Woodward Medical Research Institute for performing crystallization buffer screens. This work was supported by the National Science Foundation to AGV (NSF CAREER Award No. 1941966), a Rackham Graduate Student Research Grant to JLB, a NIH Cellular Biotechnology Training Program grant to JAB (Award No. T32 GM145304),and by research initiation funds provided by the MCDB Department to AGV.

## Additional information

### Funding

| Funder | Grant reference number | Author |
|---|---|---|
| National Science Foundation | 1941966 | Anthony G Vecchiarelli |
| National Science Foundation | 1817478 | Anthony G Vecchiarelli |
| National Institutes of Health | T32 GM145304 | Jordan A Byrne |

The funders had no role in study design, data collection, and interpretation, or the decision to submit the work for publication.

### Author contributions

Joseph L Basalla, Conceptualization, Resources, Data curation, Formal analysis, Funding acquisition, Investigation, Visualization, Methodology, Writing – original draft, Writing – review and editing; Claudia A Mak, Data curation, Formal analysis, Methodology, Writing – review and editing; Jordan A Byrne, Data curation; Maria Ghalmi, Data curation, Formal analysis; Y Hoang, Resources, Methodology, Writing – review and editing; Anthony G Vecchiarelli, Conceptualization, Formal analysis, Supervision, Funding acquisition, Investigation, Methodology, Writing – original draft, Project administration, Writing – review and editing

### Author ORCIDs

Joseph L Basalla (ID) http://orcid.org/0009-0006-5444-344X
Claudia A Mak (ID) http://orcid.org/0000-0001-5903-1766
Jordan A Byrne (ID) http://orcid.org/0000-0002-9480-6587
Y Hoang (ID) http://orcid.org/0000-0001-8804-534X
Anthony G Vecchiarelli (ID) http://orcid.org/0000-0002-6198-3245

### Decision letter and Author response

Decision letter https://doi.org/10.7554/eLife.81362.sa1
Author response https://doi.org/10.7554/eLife.81362.sa2

## Additional files

### Supplementary files
• MDAR checklist

## Data availability

All data generated or analyzed during this study are included in the manuscript and supporting file. Any materials used in this study are described in detail in Materials and methods or can be accessed by contacting the authors.

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
