## [Editor Report]

This work builds on the recent identification of the two-factor system that was discovered to be essential for the maintenance of carboxysome distribution (McdAB). McdB, a member of the two-factor system, is the focus of study here, and the intent is to uncover the driving forces for and the functional roles of phase separation. The key findings are that an N-terminal intrinsically disordered region modulates condensate solubility, a central coiled-coil dimerizing domain drives condensate formation, and oligomerization through the C-terminal domain provides the increased valency that contributes to the associative phase transitions.

---

## [Decision Letter]

**Decision letter after peer review:**

Thank you for submitting your article "Dissecting the phase separation and oligomerization activities of the carboxysome positioning protein McdB" for consideration by *eLife*. Your article has been reviewed by 3 peer reviewers, one of whom is a member of our Board of Reviewing Editors, and the evaluation has been overseen by David Ron as the Senior Editor. The following individual involved in review of your submission has agreed to reveal their identity: W. Seth Childers (Reviewer #3).

The Reviewing Editor has drafted this to help you prepare a revised submission. Please note that there are numerous issues that have come up in the evaluations provided by all three reviewers. Several key experiments and the integration of the three strands of investigation are missing and must be included for the revision to be deemed to be fully responsive to the concerns of the reviewers.

Essential revisions:

1) Reviewer 1 raises numerous specific issues pertaining to the design of the experiments, interpretations, and the potential for alternative approaches, analyses, and inferences. Please pay close attention to each of these concerns.

2) Likewise, Reviewer 2 is concerned about the inadequacy of some of the measurements, and the need for revisiting some of the analyses and inferences.

Please provide detailed, point-by-point responses to all the comments and concerns raised by all three reviewers, with particular attention to the more guarded assessments of reviewers 1 and 2.

3) It appears that there are several strong statements that should be reconsidered and either modified or eliminated. Please do so in the revised version.

*Reviewer #1 (Recommendations for the authors):*

It would be very helpful to go down a systematic investigation to set up various topologies for the dimers and hexamers and rule in or rule out the types of interactions that contribute to the oligomerization vs. networking transitions. It is likely that the sub-saturated solution is not restricted to dimers and hexamers. There likely are other oligomers, possibly even higher order ones present that contribute to the networking transition above a threshold concentration. The shape of the phase boundary obtained from concentration, [salt], and pH titrations is of significant import for understanding the driving forces for phase separation. The use of crowders, especially PEG, is going to lead to confounding interpretations, that are best avoided by judicious use of a range of deletion constructs, or better yet, systematic mutagenesis directed toward the full-length protein. At this juncture, there is considerable circumstantial evidence pointing to a contribution of electrostatic interactions between the N- and C-termini and how this occurs in cis vs. trans is what needs resolution.

*Reviewer #2 (Recommendations for the authors):*

1. Figure 1B: A potential reason for the discrepancy between i-TASSER prediction of a helical region in the n-terminal and your trypsin assay is that small fragments might be too small to be identified on a gel. Please make this point clear.

2. The cell images in Figure 6 suggest that mCh-McdB[-3] is not expressed in most cells. Is the analysis of proportions of cells with foci taken from the entire population or just cells that are expressing mCherry? Please provide details on your image analysis.

3. It is difficult to interpret the graphical representation of the mutant in Figure 5B. Please add a label next to each graphic.

4. In the discussion, the authors write Lines 415-156: "it is attractive to speculate that coating carboxysomes with a viscous, pH-sensitive McdB phase may influence the diffusivity of carboxysome shells to protons." If McdB coats carboxysomes, it would be useful to comment on 2D vs. 3D phase separation.

*Reviewer #3 (Recommendations for the authors):*

(1) In the modeling (Figure 7), the authors proposed that the N-terminus of McdA functions as a MORF. This has the potential to open up an exciting avenue of study. However, the experimental support is underdeveloped. I would recommend either removing it from this study or developing it with more experimental support if the authors prefer to keep it in this study.

a. For example, the authors could leverage their purified McdB glutamine variants and δ-IDR variant for in vitro McdA-McdB binding assays.

b. A MORF requires a change from unstructured to structured, and this data may be harder to obtain. However, the authors would be in a stronger position to speculate it's a MORF with in vitro binding data (or reference it if published elsewhere).

c. If the N-IDR regulates both phase separation of McdB and interacts with McdA, it raises intriguing questions the authors suggest in their discussion. Does McdA enhance or diminish McdB's phase separation? How might this fit into the in vivo model of McdA-McdB function in the spatial control of carboxysomes (McdA regulates McdB phase separation versus just being a client that does not impact phase separation)?

(2) What is the measured or estimated copy number of McdB in cells? How does that compare with the observed Csat of McdB? If it significantly differs, it may imply that other proteins (McdA or unknown clients) impact/regulate McdB's phase separation.

(3) To compare circular dichroism spectra of different McdB constructs or estimate secondary structure content, it's best to normalize the raw millidegrees data to mean residue molar ellipticity (Figure 1A, 2B, S5).

(4) In Figure 6, to bolster the conclusions of the SDS-PAGE gel showing increased expression at longer times, the authors should consider analyzing their existing microscopy data (Figure 6A) on the single cell level. A plot of foci intensity vs. total cell intensity would provide further evidence that the assemblies are concentration-dependent and account for cell-to-cell heterogeneity in McdB levels.

[Editors’ note: further revisions were suggested prior to acceptance, as described below.]

Thank you for resubmitting your work entitled "Dissecting the phase separation and oligomerization activities of the carboxysome positioning protein McdB" for further consideration by *eLife*. Your revised article has been evaluated by David Ron (Senior Editor) and a Reviewing Editor.

The manuscript has been improved but we find that the discussion should be revised to better reflect the main findings and to be constrained by the findings. Reviewer three made specific proposals for revisions that were endorsed in the editorial consultation. Since, no additional data will be required, we expect the requested revisions to be undertaken efficiently and look forward to a revised version, fit for publication.

Overall, the study substantially improved the manuscript during this revision, addressing many of the reviewer's concerns. These major improvements include:

1) The authors provide a new study in Figure 8, examining how and McdB[-3] variant that does not phase separate or interact with McdA impacts carboxysome homeostasis. Given that this variant impacts both phase separation and McdA binding, the effect of McdB[-3] was examined in a McdA null background. From these experiments, they conclude that phase separation contributes to its association with carboxysomes and influences carboxysome enzyme content.

2) At Reviewer 1's suggestion, the authors implemented a dynamic light scattering study that suggested the presence of higher-ordered intermediated below csat in a pH-dependent manner. This new preliminary data allows the authors to discuss the role of networking and percolation. Additionally, the authors examined the time-dependent viscoelastic properties of McdB. This allowed the authors to discuss the phenomena more deeply than simply LLPS.

3) The authors more carefully studied csat and observed small condensates at 2 µM. This brings some confidence that concentrations may be low enough to drive phase separation in vivo, albeit higher than the authors' estimate of csat in cells.

4) The authors removed some of the weaker parts of the text: The q-rich assignment, the study of a SLIM. This critically leaves behind a more robust core of the paper for consideration.

Some issues pointed out by the reviewers are unresolved:

1) Orientation of the dimers,

2) Topology of the hexamer,

3) Reliance on PEG inducers for deletion constructs, but not full-length,

4) N- and C-termini interactions as cis or trans. While unanswered, the authors altered the scope to consider the consequences of variants in vivo.

Overall, I remain enthusiastic about this study. The study represented a substantial scope of work that would interest the carboxysome and bacterial cell biology communities. However, based on adding new experiments in Figure 8, I have some additional questions for the authors to consider.

1) One potential conclusion from Figure 8 is that the interaction with McdA and phase separation are not required for foci formation in vivo. This is discussed a bit in lines 357-359. However, the reader could benefit from being discussed more directly that foci formation does not require phase separation. The subcellular localization mechanism, at least for McdB[-3], seems to occur through carboxysome binding and is consistent with the authors past two-hybrid assays. So, in this case we cannot use foci formation in vivo as evidence of McdB phase separation. The new experiments in Figure 8 were overall impactful and raised several new questions:

a. In the discussion, the authors claim evidence of in vivo condensate formation. However, I think Figure 8 may provide some evidence to counter that model. If McdB formed a condensate, one prediction might be that the McdB[-3] variant that cannot phase separate might increase the dilute phase soluble fraction. In contrast to this prediction, the authors found that variants with reduced phase separation in vitro caused a substantial decrease in the dilute phase concentration in vivo (Figure 8). Can the authors explain their model for this unexpected finding and rationalize how it is consistent with phase separation in vivo? My only thought was that when the McdB phase separates, it cannot fully occupy all carboxysome binding sites. If you eliminate the phase separation of McdB, then McdB can occupy the carboxysome binding sites leading to increased foci intensity.

b. As an alternative model might, the region mutated in McdB[-3] regulate three distinct functions: phase separation, McdA binding, and negative regulation of carboxysome binding. Mutation could enhance carboxysome binding if this region also negatively regulates carboxysome binding. Perhaps this is connected to the "polyampholytic properties between their N- and C-termini" pointed out by the authors. To address this potential, can the authors discuss what is known about how McdB binds to carboxysomes? Can the authors provide or point out any evidence to rule out this model? If not, it may be worth discussing this as an alternative model.

c. I'd encourage the authors to be more cautious in interpreting condensate formation in vivo (Lines 407,431,478). It's a clear and robust possibility in Synechococcus elongatus but not rigorously tested yet. For ParB, I felt the combination of studies in Ptacin et al. PNAS 2014, Guilhas et al. Mol Cell 2020, and Babl et al. JBC 2022 provided good minimal support for in vivo condensate formation of ParB. This may require a larger scope of work: in vitro FRAP, single molecular tracking, hexanediol/lipoic acid experiments, etc. Is the subcellular localization mechanism different for McdB and McdB[+3]? Perhaps superresolution imaging could distinguish the McdB coating of the carboxysome versus a larger condensate bound to the carboxysome (experiments similar to Ptacin et al. PNAS 2014 superresolution imaging in Figure 4B) if McdB and McdB[+3] displayed protein assemblies adjacent to the carboxysome that might support in vivo phase separation of McdB. In contrast, if both assemblies appear to colocalize with the carboxysome surface uniformly, that might suggest McdB coats the carboxysome surface (potentially still as a 2D-constrained condensate). Given this enormous and challenging scope, perhaps a section on the challenges of annotating in vivo McdB condensates would be helpful to the reader. A description of approaches the authors would like to see to make a definitive assignment.

2) The impact on carboxysome loading was an interesting observation. Are there other potential reasons to explain why McdB-3 impacts carboxysome loading? Does phase separation facilitate the loading and packaging of the enzymes? Is it possible that McdB-3 impacts the protein expression levels of RbcS-mTQ, which is the root cause of reduced RbcS-mTQ foci intensity in carboxysomes? Can the authors provide a total cell intensity analysis of RbcS-mTQ for the studied strains to address this possibility?

---

## [Author Response]

Essential revisions:1) Reviewer 1 raises numerous specific issues pertaining to the design of the experiments, interpretations, and the potential for alternative approaches, analyses, and inferences. Please pay close attention to each of these concerns.2) Likewise, Reviewer 2 is concerned about the inadequacy of some of the measurements, and the need for revisiting some of the analyses and inferences.Please provide detailed, point-by-point responses to all the comments and concerns raised by all three reviewers, with particular attention to the more guarded assessments of reviewers 1 and 2.3) It appears that there are several strong statements that should be reconsidered and either modified or eliminated. Please do so in the revised version.

We have performed a number of additional experiments aimed at addressing issues raised by the reviewers. The additional data and revisions significantly improve the findings, interpretation, conclusions, and general impact of the paper. Ultimately, the additional data also required us to somewhat reframe the focus of the paper. Please see details below in our point-by-point response to reviewers. We thank the reviewers for their time and thoughtful consideration of our work.*Reviewer #1 (Recommendations for the authors):*

It would be very helpful to go down a systematic investigation to set up various topologies for the dimers and hexamers and rule in or rule out the types of interactions that contribute to the oligomerization vs. networking transitions. It is likely that the sub-saturated solution is not restricted to dimers and hexamers. There likely are other oligomers, possibly even higher order ones present that contribute to the networking transition above a threshold concentration.

Thank you for this insight. We now provide dynamic light-scattering data to determine size distributions of McdB in sub-saturated solutions (Figure 3). The reviewer was correct. We observed the presence of discreet higher-order species in these solutions that were pH-dependent. The data further adds to our model of McdB condensate formation and has significantly strengthened the manuscript.

The shape of the phase boundary obtained from concentration, [salt], and pH titrations is of significant import for understanding the driving forces for phase separation.

These phase diagrams are included (Figure S5) and did indeed help us identify basic residues as key factors for condensate formation (Figures 5, 6).

The use of crowders, especially PEG, is going to lead to confounding interpretations, that are best avoided by judicious use of a range of deletion constructs, or better yet, systematic mutagenesis directed toward the full-length protein.

We have included data showing that the type of crowding agent, when used, did not notably impact McdB condensate behavior (Figure S4). Together, the data shows that our proposed model of McdB condensate formation was not dependent on the presence of a specific crowder. We performed assays on both deletion constructs as well as substitution mutants on the full-length protein to assess McdB condensate behavior (Figures 4-6).

At this juncture, there is considerable circumstantial evidence pointing to a contribution of electrostatic interactions between the N- and C-termini and how this occurs in cis vs. trans is what needs resolution.

Thank you. We have attempted to address this as described in response #6. We have added a section in the discussion that highlights this specific issue: “*McdB homologs have polyampholytic properties between their N- and C-termini*”

Reviewer #2 (Recommendations for the authors):1. Figure 1B: A potential reason for the discrepancy between i-TASSER prediction of a helical region in the n-terminal and your trypsin assay is that small fragments might be too small to be identified on a gel. Please make this point clear.

Based off Reviewer 1 and 3 comments, we have entirely removed our focus on the small discrepancies between these results and the relevant MoRF analyses.

2. The cell images in Figure 6 suggest that mCh-McdB[-3] is not expressed in most cells. Is the analysis of proportions of cells with foci taken from the entire population or just cells that are expressing mCherry? Please provide details on your image analysis.

After the onset of induction, the majority of cells show mCherry signal. The analysis was done on the entire population of cells, and protein levels were comparable between all strains as indicated in Figure 7B.

3. It is difficult to interpret the graphical representation of the mutant in Figure 5B. Please add a label next to each graphic.

Apologies. We now include labels.

4. In the discussion, the authors write Lines 415-156: "it is attractive to speculate that coating carboxysomes with a viscous, pH-sensitive McdB phase may influence the diffusivity of carboxysome shells to protons." If McdB coats carboxysomes, it would be useful to comment on 2D vs. 3D phase separation.

Thank you for this insight. It is very possible this is a contributing factor. We have added a mention to it in the Discussion section titled “*pH as a potential underlying regulator for McdB condensate solubility and its association with carboxysomes*”.

Reviewer #3 (Recommendations for the authors):1) In the modeling (Figure 7), the authors proposed that the N-terminus of McdA functions as a MORF. This has the potential to open up an exciting avenue of study. However, the experimental support is underdeveloped. I would recommend either removing it from this study or developing it with more experimental support if the authors prefer to keep it in this study.a. For example, the authors could leverage their purified McdB glutamine variants and δ-IDR variant for in vitro McdA-McdB binding assays.b. A MORF requires a change from unstructured to structured, and this data may be harder to obtain. However, the authors would be in a stronger position to speculate it's a MORF with in vitro binding data (or reference it if published elsewhere).c. If the N-IDR regulates both phase separation of McdB and interacts with McdA, it raises intriguing questions the authors suggest in their discussion. Does McdA enhance or diminish McdB's phase separation? How might this fit into the in vivo model of McdA-McdB function in the spatial control of carboxysomes (McdA regulates McdB phase separation versus just being a client that does not impact phase separation)?

The MoRF analyses has been removed, and will be coupled to another study in the lab focused on McdB interactions with McdA.

2) What is the measured or estimated copy number of McdB in cells? How does that compare with the observed Csat of McdB? If it significantly differs, it may imply that other proteins (McdA or unknown clients) impact/regulate McdB's phase separation.

We do not know the in vivo concentration of McdB. We have tried several antibodies against McdB, and a few were good enough to detect the presence of McdB, but not quantifiably. We therefore believe in vivo McdB levels are low (sub-μM), and definitely lower than the range we previously used in our in vitro studies. We now include a titration of McdB at lower concentrations than we had in the previous version of our report (Figure 3). We see that McdB can form condensates in vitro at concentrations lower than 2 µM, and in the presence or absence of different crowding agents (Figure S4) and across a range of buffering conditions (Figures 5, S5). Certainly, its interactions with other proteins in vivo will affect its phase behavior, but the robust formation of condensates in vitro shows McdB can form them under many physiologically relevant conditions. Lastly, a major focus of the current report was to identify mutations in vitro that specifically affected condensate formation and not structure or oligomerization, then observe these in vivo, which we have included (Figure 8). The identification of phenotypes specifically linked to McdB condensate solubility suggests that McdB may utilize phase separation in vivo, regardless of the concentrations we used to identify mutation sets in vitro.

3) To compare circular dichroism spectra of different McdB constructs or estimate secondary structure content, it's best to normalize the raw millidegrees data to mean residue molar ellipticity (Figure 1A, 2B, S5).

Thank you for the suggestion. We have normalized the spectra in Figures 2 and S3 where we are comparing constructs of significantly different size and weight. The comparisons yield the same conclusions.

4) In Figure 6, to bolster the conclusions of the SDS-PAGE gel showing increased expression at longer times, the authors should consider analyzing their existing microscopy data (Figure 6A) on the single cell level. A plot of foci intensity vs. total cell intensity would provide further evidence that the assemblies are concentration-dependent and account for cell-to-cell heterogeneity in McdB levels.

We believe the microscopy combined with the quantification of the proportion of cells with foci (Figure 7AB) and the SDS-PAGE quantification (Figure 7CD) is good evidence towards our conclusions. The error bars in these quantifications (Figure B and D) also account for the cell-to-cell heterogeneity in these analyses. The issue with a plot of foci intensity versus total cell intensity is that the majority of cells, particularly at low McdB concentration early on in the induction, have no foci at all. Therefore this type of analysis skews the data towards the fraction of the cell population containing foci. In addition, because McdB has a significant cytoplasmic fraction even when forming a focus, it if very difficult for MicrobeJ and other imaging analysis programs to differentiate between focus and cytoplasm; thus further introducing error in accurately distinguishing the two populations. The quantifications we currently provide are unbiased and population based. However, we agree this is a useful quantification when foci are always present and significantly brighter than the cytoplasmic fraction. We use the suggested analysis in Figure 8 where we look at McdB foci formation in *S. elongatus* cells. Thank you for the excellent suggestion.

[Editors’ note: what follows is the authors’ response to the second round of review.]

1) One potential conclusion from Figure 8 is that the interaction with McdA and phase separation are not required for foci formation in vivo. This is discussed a bit in lines 357-359. However, the reader could benefit from being discussed more directly that foci formation does not require phase separation. The subcellular localization mechanism, at least for McdB[-3], seems to occur through carboxysome binding and is consistent with the authors past two-hybrid assays. So, in this case we cannot use foci formation in vivo as evidence of McdB phase separation. The new experiments in Figure 8 were overall impactful and raised several new questions:a. In the discussion, the authors claim evidence of in vivo condensate formation. However, I think Figure 8 may provide some evidence to counter that model. If McdB formed a condensate, one prediction might be that the McdB[-3] variant that cannot phase separate might increase the dilute phase soluble fraction. In contrast to this prediction, the authors found that variants with reduced phase separation in vitro caused a substantial decrease in the dilute phase concentration in vivo (Figure 8). Can the authors explain their model for this unexpected finding and rationalize how it is consistent with phase separation in vivo? My only thought was that when the McdB phase separates, it cannot fully occupy all carboxysome binding sites. If you eliminate the phase separation of McdB, then McdB can occupy the carboxysome binding sites leading to increased foci intensity.

Thank you for these thoughtful considerations. We agree that the in vivo results are complex and should certainly be discussed further. As suggested, it is possible that when McdB self-associates via phase separation, it is less available to occupy carboxysome binding sites. This is especially true if McdB is the limiting factor compared to carboxysome sites, which we assume to be true. We have added a new Discussion section titled “Considerations for McdB condensate formation in vivo” where this is discussed.

b. As an alternative model might, the region mutated in McdB[-3] regulate three distinct functions: phase separation, McdA binding, and negative regulation of carboxysome binding. Mutation could enhance carboxysome binding if this region also negatively regulates carboxysome binding. Perhaps this is connected to the "polyampholytic properties between their N- and C-termini" pointed out by the authors. To address this potential, can the authors discuss what is known about how McdB binds to carboxysomes? Can the authors provide or point out any evidence to rule out this model? If not, it may be worth discussing this as an alternative model.

Thank you. This is related to the model we describe in the point above. We speculate that the C-terminus of McdB is involved in self-association with the N-terminus via polyampholytic properties and also contains regions that may be responsible for interactions with carboxysomes. In this way, McdB self-association may compete against carboxysome binding and therefore negatively regulate it. We discuss this in the new Discussion section titled “Considerations for McdB condensate formation in vivo”.

c. I'd encourage the authors to be more cautious in interpreting condensate formation in vivo (Lines 407,431,478). It's a clear and robust possibility in Synechococcus elongatus but not rigorously tested yet. For ParB, I felt the combination of studies in Ptacin et al. PNAS 2014, Guilhas et al. Mol Cell 2020, and Babl et al. JBC 2022 provided good minimal support for in vivo condensate formation of ParB. This may require a larger scope of work: in vitro FRAP, single molecular tracking, hexanediol/lipoic acid experiments, etc. Is the subcellular localization mechanism different for McdB and McdB[+3]? Perhaps superresolution imaging could distinguish the McdB coating of the carboxysome versus a larger condensate bound to the carboxysome (experiments similar to Ptacin et al. PNAS 2014 superresolution imaging in Figure 4B) if McdB and McdB[+3] displayed protein assemblies adjacent to the carboxysome that might support in vivo phase separation of McdB. In contrast, if both assemblies appear to colocalize with the carboxysome surface uniformly, that might suggest McdB coats the carboxysome surface (potentially still as a 2D-constrained condensate). Given this enormous and challenging scope, perhaps a section on the challenges of annotating in vivo McdB condensates would be helpful to the reader. A description of approaches the authors would like to see to make a definitive assignment.

Thank you. It is certainly true that more studies are required to give a robust model on how McdB condensate formation is playing a role in vivo, and super-resolution techniques would of course be useful. We now mention this in the Discussion section “Considerations for McdB condensate formation in vivo” and also mention that while our biochemical approach was useful, it is only an initial step in developing the model.

2) The impact on carboxysome loading was an interesting observation. Are there other potential reasons to explain why McdB-3 impacts carboxysome loading? Does phase separation facilitate the loading and packaging of the enzymes? Is it possible that McdB-3 impacts the protein expression levels of RbcS-mTQ, which is the root cause of reduced RbcS-mTQ foci intensity in carboxysomes? Can the authors provide a total cell intensity analysis of RbcS-mTQ for the studied strains to address this possibility?

Thank you. One thing we failed to mention is that McdB not only interacts with carboxysome shells, but can also interact with a carboxysome core component, notably, CcmM. CcmM is the scaffolding protein that phase separates with Rubisco, and is thought to facilitate Rubisco loading into carboxysomes. Thus, it is possible that McdB phase separation affects Rubisco loading through interactions with CcmM. Starting on Line 495, We have added mention to this in the Discussion section “pH as a potential underlying regulator for McdB condensate solubility and its association with carboxysomes”:

“Future studies will determine the nature of McdB association with carboxysomes, and how condensate formation influences this association. For example, we have shown that McdB strongly associates with carboxysome shell proteins via bacterial two-hybrid assays (7). It is attractive to speculate that McdB undergoes pre-wetting interactions with the 2D surface of the carboxysome shell, which then nucleates surface-assisted condensation. Such 2D interactions would significantly impact McdB phase boundaries (66). It is also possible that McdB phase separation directly modulates carboxysome fluidity and Rubisco content. Carboxysomes were traditionally thought of as paracrystalline, but recent data in S. elongatus shows that carboxysome biogenesis begins with Rubisco forming a condensate with its linker protein CcmM (67). Intriguingly, our bacterial-two hybrid assays have shown that, in addition to shell proteins, McdB also interacts strongly with CcmM. It is therefore possible that McdB and its phase separation activity influences carboxysome composition, fluidity, and function through interactions with the Rubisco-CcmM condensate. Such a model helps explain defects in Rubisco loading in our condensation-deficient McdB stain. Teasing apart the stable protein-protein interactions between McdB and carboxysomes from the dynamic processes governing condensate solubility will therefore be of significant importance.”